# Development of a unit-based industrial emission inventory in the Beijing-Tianjin-Hebei region and resulting improvement in air quality modelling

Haotian Zheng[a,b] [1], Siyi Cai[a], [1], Shuxiao Wang[a,b*], Bin Zhao[c*], Xing Chang[a,b], Jiming Hao[a,b]

[a] State Key Joint Laboratory of Environmental Simulation and Pollution Control, School of Environment, Tsinghua University, Beijing, 100084, China
[b] State Environmental Protection Key Laboratory of Sources and Control of Air Pollution Complex, Beijing 100084, China
[c] Joint Institute for Regional Earth System Science and Engineering and Department of Atmospheric and Oceanic Sciences, University of California, Los Angeles, CA 90095, USA

*Correspondence to*: S. Wang (shxwang@tsinghua.edu.cn) and B. Zhao (zhaob1206@ucla.edu)

[1] These authors contributed equally to this study.

**Abstract.**

The Beijing-Tianjin-Hebei (BTH) region is a metropolitan area with the most severe fine particle ($PM_{2.5}$) pollution in China. Accurate emission inventory plays an important role in air pollution control policy making. In this study, we develop a unit-based emission inventory for industrial sectors in the BTH region, including power plants, industrial boilers, steel, non-ferrous metal, coking, cement, glass, brick, lime, ceramics, refinery, and chemical industries, based on detailed information for each enterprise, such as location, annual production, production technology/process and air pollution control facilities. In the BTH region, the emissions of sulfur dioxide ($SO_2$), nitrogen oxide ($NO_x$), particulate matter with diameter less than 10 μm ($PM_{10}$), $PM_{2.5}$, black carbon (BC), organic carbon (OC), and non-methane volatile organic compounds (NMVOCs) from industrial sectors are 869 kt, 1164 kt, 910 kt, 622 kt, 71 kt, 63 kt and 1390 kt in 2014, respectively, accounting for 61%, 55%, 62%, 56%, 58%, 22% and 36%, respectively, of the total emissions. Compared with the traditional proxy-based emission inventory, much less emissions in the high-resolution unit-based inventory are allocated to the urban center because of the accurate positioning of industrial enterprises. We apply the Community Multi-scale Air Quality (CMAQ) model

simulation to evaluate the unit-based inventory. The simulation results show that the unit-based emission inventory gives better performance of both PM$_{2.5}$ and gaseous pollutants than the proxy-based emission inventory. The normalized mean biases (NMBs) are 81%, 21%, 1% and -7% for concentrations of SO$_2$, NO$_2$, ozone and PM$_{2.5}$, respectively, with the unit-based inventory, in contrast to 124%, 39%, -8% and 9%

with the proxy-based inventory. Furthermore, the concentration gradients of PM$_{2.5}$, which are defined as the ratio of urban concentration to suburban concentration, are 1.6, 2.1 and 1.5 in January and 1.3, 1.5 and 1.3 in July, for simulations with the unit-based inventory, simulations with the proxy-based inventory, and observations, respectively, in Beijing. For ozone, the corresponding gradients are 0.7, 0.5 and 0.9 in January and 0.9, 0.8 and 1.1 in July, implying that the unit-based emission inventory better reproduces

the distributions of pollutant emissions between the urban and suburban areas.

## 1 Introduction

The Beijing-Tianjin-Hebei (BTH) region is the political, economic and cultural center of China. According    to    China    National    Environmental    Monitoring    Centre    (data    source: http://106.37.208.233:20035/), in 2017, the annual average concentration of PM$_{2.5}$ in Beijing, Tianjin and

Hebei are 65.6, 63.8 and 57.1 μg/$m^3$, ranking second, third and sixth among all provinces. The severe PM$_{2.5}$ pollution in the BTH region is largely attributed to the substantial emissions of air pollutants (Zhao et al., 2017a). An accurate emission inventory, in terms of both emission rates and spatial distribution, is imperative for an adequate understanding of the sources and formation mechanism of the serious air pollution.

The spatial distribution is one of the most uncertain component of emission inventories considering the diverse source categories and complex emission characteristics. The traditional method of spatial allocation is to distribute the emissions by administrative region into grids based on spatial proxies such as population, gross domestic product (GDP), road map, land use data and nighttime lights (Geng et al., 2017;Oda and Maksyutov, 2011;Streets et al., 2003). The results may deviate significantly from the actual

spatial distributions of many sources (Zhou and Gurney, 2011), especially the power and industrial

sources, which contribute over 50% of the total $PM_{2.5}$ emissions in China (Zhao et al., 2013a). Due to the stricter air quality regulation and higher land price in urban area, people tend to build factories in suburban area where the population density and GDP are lower. Zheng et al. (2017) studied the influence of the resolution of gridded emission inventory and found that there were large biases when the inventory was

distributed to very fine resolution following the traditional proxy-based allocation method. The emission inventory could be significantly improved with detailed information of point sources such as power plants, steel plants, cement plants, etc. The high spatial resolution of the inventory may subsequently improve the air quality modelling results and enable a better source apportionment of air pollution (Zhao et al., 2017c).

A couple of studies have developed the emission inventory in the BTH region (Li et al., 2017;Wang et al., 2014), and some others have provided emission estimates for this region as part of national or larger-scale emission inventories (Ohara et al., 2007;Stohl et al., 2015). However, only limited studies estimated the emissions by individual point sources (i.e., unit-based emission inventory). Zhao et al. (2008), Chen et al. (2014) and Liu et al. (2015) established unit-based emission inventories of coal-fired power plants

in China. Wang et al. (2016b) and Wu et al. (2015) developed an emission inventory of steel industry. Lei et al. (2011) and Chen et al. (2015) established an emission inventory of cement industry in China. Qi et al. (2017) established an emission inventory in BTH region with power and major industrial sources treated as point sources. These studies usually focused on one or several major industries, and did not cover all industrial sectors in the BTH region. Moreover, these previous studies seldom validated the unit-

based emission inventory or evaluated the improvement it brings to air quality simulation.

In this study, we developed a unit-based emission inventory of industrial sectors for the Beijing-Tianjin-Hebei region. A three-domain nested simulation by WRF-CMAQ model was applied to evaluate the emission inventory. In order to study the influence of the point sources, we compared the simulation results of this emission inventory with those of a traditional proxy-based emission inventory.

## 2 Materials and methods

### 2.1 High-resolution emission inventory for Beijing-Tianjin-Hebei region

A unit-based method is applied to quantify the emissions from industrial sectors such as power plant, industrial boiler, iron and steel production, non-ferrous metal smelter, coking, cement, glass, brick, lime, ceramics, refinery, and chemical industries in 2014. The product yields used for estimating emissions of each sector are shown in Table S4. The pollutant emissions from each industrial enterprise are calculated from activity level (energy consumption for power plants and industrial boilers, and product yield for other sectors), emission factor, and removal efficiency of control technology, as shown in the following equation:

$$E_{i,j} = A_j \times EF_{i,j} \times \left(1 - \eta_{i,j}\right) \tag{1}$$

where $E_{i,j}$ is emissions of pollutant i from industrial enterprise j, $A_j$ is activity level of industrial enterprise j, $EF_{i,j}$ is uncontrolled emission factor of pollutant i from industrial enterprise j, and $\eta_{i,j}$ is removal efficiency of pollutant i by control technology in enterprise j. $\eta_{i,j}$ is determined by the production process and control technology of the industrial enterprise. The $EF_{i,j}$, which depends on the production process of the industrial enterprise, are calculated according to the sulfur and ash contents of fuels (e.g. coal) used in each province (for PM and $SO_2$), or obtained from our previous study (Zhao et al., 2013b) (for other pollutants).

Some industrial sources involve multiple production process, such as iron and steel production and cement production. Taking cement production for example, emissions are calculated by using the following equation:

$$E_{i,j} = \sum_m \left(AK_{j,m} \times EF_{i,m} \times \left(1 - \eta_{i,j,m}\right)\right) + \left(AC_j \times ef_i \times \left(1 - \eta_{i,j}\right)\right) \tag{2}$$

where $E_{i,j}$ is emissions of pollutant i from industrial enterprise j, $AK_{j,m}$ is the amount of clinker produced by the clinker burning process m of the enterprise j, $EF_{i,m}$ is uncontrolled emission factor for pollutant i from the clinker burning process m, $\eta_{i,j,m}$ is removal efficiency of pollutant i from the clinker burning process m in enterprise j, $AC_j$ is the amount of cement produced by enterprise j, $ef_i$ is uncontrolled

emission factors from the clinker processing stage ($ef_i$=0 if i is not particulate matter), $\eta_{i,j}$ is removal efficiency of pollutant i in enterprise j. $\eta_{i,j,m}$ and $\eta_{i,j}$ both depend on the control technology of the industrial enterprise.

The production processes represented by the first and second terms of equation (2) are frequently performed in different enterprises. For example, for cement production, clinker may be produced in one enterprise and subsequently processed in another enterprise, which is very common. For each enterprise, we calculate the emission of each production process. Specifically, the total emission of enterprise j is the sum of the emissions of all production processes in that enterprise. If processes are divided to multiple enterprises, the emission will be considered in the calculation of the emission of each individual enterprise.

For all power and industrial sources except industrial boilers, we collect their detailed information, including latitude/longitude, annual product, production technology/process, and pollution control facilities from compilation of power industry statistics (China Electricity Council, 2015b), China Iron and Steel Industry Association (http://www.chinaisa.org.cn), China Cement Association (http://www.chinacca.org), Chinese environmental statistics (collected from provincial environmental protection bureaus), the first national census of pollution sources (National Bureau of Statistics (NBS), 2010) and bulletin of desulfurization and denitrification facilities from Ministry of Ecology and Environment of China (http://www.mee.gov.cn). These emission sources include 242 power plants, 333 iron and steel plants, 639 cement plants, 151 nonferrous metal smelters, 211 lime plants, 1222 brick and tile plants, 37 ceramic plants, 42 glass plants, 106 coking plants, 21 refinery plants, and 328 chemical plants. The iron and cement sectors are divided to specific industrial processes. For industrial boilers, we obtained the location, fuel use amount, and control technologies of over 8 thousand industrial boilers in Beijing, Tianjin, and Hebei from Xue et al. (2016), Tianjin Environmental Protection Bureau, and Hebei Environmental Protection Bureau.

Plume rise is caused by buoyancy effect and momentum rise (Briggs, 1982). Therefore, the stack information including stack height, flue gas temperature, chimney diameter and flue gas velocity is essential for plume rise calculation. For power plants, we get the stack height from Compilation of power industry statistics (China Electricity Council, 2015b). For the stack height of cement factories, we refer

to the emission standard of air pollutants for cement industry (Ministry of Environmental Protection of China, 2013). For the stack height of glass, brick, lime and ceramics industries, we refer to emission standard of air pollutants for industrial kiln and furnace (Ministry of Environmental Protection of China, 1997). For the stack height of non-ferrous metal smelter, coking, refinery and chemical industries, as well as the flue gas temperature, chimney diameter and flue gas velocity for all industrial sectors, we refer to the national information platform of pollutant discharge permit (http://114.251.10.126/permitExt/outside/default.jsp), where we can find very detailed information of the plants with the pollutant discharge permit. For the sources without the pollutant discharge permit, we use the parameters of the plant with a similar production output or coal consumption. Individual information of stacks is applied to each production process. The locations of different processes in the same enterprise are usually assumed to be the same.

The emission inventory for other sources, including residential sources, transportation, solvent use, and open burning, is developed based on the "top-down method" following our previous work (Fu et al., 2013;Wang et al., 2014;Zhao et al., 2013b). The method is the same as **Eq (1)** except that the emissions are calculated for individual prefecture-level city rather than individual enterprise. The activity data and technology distribution for each sector are derived based on the Statistics Yearbook (Beijing Municipal Bureau of Statistics, 2015;Hebei Municipal Bureau of Statistics, 2015;National Bureau of Statistics (NBS), 2015h, g, f, e, i, j, a, b, c, d;Tianjin Municipal Bureau of Statistics, 2015), a wide variety of Chinese technology reports (China Electricity Council, 2015a;National Bureau of Statistics (NBS), 2012), and an energy demand modelling approach. **Fig.S1** shows energy consumption in the BTH region in 2014. We compared the sum of the energy consumption for each plant with the energy statistics. The sum of individual plants accounts for over 90% of the energy consumption or product yield reported in the statistics. For the plants not included in the preceding data sources, we calculate the emission by using "top-down method". The emission factors are also obtained from Zhao et al. (2013b). The speciation of $PM_{2.5}$ in both inventories is from Fu et al. (2013) while the speciation of NMVOCs is updated by Wu et al. (2017). The penetrations of removal technologies are obtained from the evolution of emission standards and a variety of technical reports (Chinese State Council, 2013).

## 2.2 Air quality model configuration

In this work, we use CMAQ version 5.0.2 (EPA, 2014) to simulate the concentration of pollutants. A three-domain nested simulation is established as shown in **Fig. 1** (left). The first domain covers almost entire area of China, Korea, Japan, and parts of India and Southeast Asia with a horizontal grid resolution of 36 km × 36 km. The second domain covers eastern China with a resolution of 12 km × 12 km. The third domain with a horizontal resolution of 4 km × 4 km focuses on the Beijing-Tianjin-Hebei region. The observational sites in Beijing-Tianjin-Hebei region are marked in **Fig. 1** (right). All of the grids are divided to 14 layers vertically from surface to an altitude of about 19 km above the ground and the thickness of the first layer is about 40 m.

In order to minimize the influence of initial condition, we choose 5 days of spin-up period. The Carbon Bond 05 (CB05) and AERO6 (Sarwar et al., 2011) are chosen as the gas-phase and aerosol chemical mechanisms, respectively. The simulation periods are January and July of 2014, representing winter and summer, respectively.

We use the Weather Research and Forecasting (WRF) model version 3.7.1 (Skamarock et al., 2008) to simulate the meteorological fields. The physics options for the WRF simulation are the Kain-Fritsch cumulus scheme (Kain, 2004), the Morrison double-moment scheme for cloud microphysics (Morrison et al., 2005), the Pleim-Xiu land surface model (Xiu and Pleim, 2001), Pleim-Xiu surface layer scheme (Pleim, 2006), ACM2 (Pleim) boundary layer parameterization (Pleim, 2007), and Rapid Radiative Transfer Model for GCMs radiation scheme (Mlawer et al., 1997). The meteorological initial and boundary conditions are generated from the Final Operational Global Analysis data (ds083.2) of the National Center for Environmental Prediction (NCEP) at a 1.0º × 1.0º and 6-h resolutions. Default profile data is used for chemical initial and boundary conditions. The Meteorology Chemistry Interface Processor (MCIP) version 4.1 is applied to process the meteorological data into a format required by CMAQ. The simulated wind speed, wind direction, temperature and humidity agree well with the observation data from the National Climate Data Center (NCDC), as detailed in the Supplementary Information.

In order to evaluate the high-resolution emission inventory with unit-based industrial sources, we developed a traditional proxy-based emission inventory with the same amount of emissions and compare the simulation results of these two emission inventories. In the proxy-based emission inventory, all sectors are allocated as area sources using spatial proxies such as population, GDP, road map and land use data. The proxies used for each sector is described in detail in Table S2. For the plants not included in the preceding data sources, it is allocated the same as proxy-based emission inventory. In order to separate the influences of horizontal and vertical distributions of emission, we developed another unit-based inventory with emission heights the same as the proxy-based inventory. In short, we call it hypo unit-based inventory. The anthropogenic emission inventory in other provinces of China was developed in our previous studies (Wang et al., 2014;Zhao et al., 2018). The emissions outside China are obtained from the MIX emission inventory (Li et al., 2017) for 2010, which is the latest year available. In the simulation with the unit-based inventory, plume rise is calculated with the built-in algorithm in CMAQ. Meteorological data are used to calculate the plume rise for all point sources. Then, the plume is distributed into the vertical layers that the plume intersects based on the pressure in each layer.

## 3 Results and discussion

### 3.1 Air pollutant emissions in Beijing-Tianjin-Hebei region

In the BTH region, the emissions of sulfur dioxide ($SO_2$), nitrogen oxide ($NO_x$), $PM_{10}$, $PM_{2.5}$, black carbon (BC), organic carbon (OC), non-methane volatile organic compounds (NMVOCs) and ammonia ($NH_3$) are 1417 kt, 2100 kt, 1479 kt, 1106 kt, 213 kt, 289 kt, 2381 kt, and 712 kt in 2014, respectively. **Fig. 2** shows the sectoral emissions for major pollutants in the BTH region by city. **Fig.S2** shows the NMVOCs speciation by sector. The emission estimates are compared with previous studies in **Fig.S3**. **Fig. 3** shows the locations and emissions of power and industrial sources.

Power plants account for 13%, 16%, and 4% of the total $SO_2$, $NO_x$, and $PM_{2.5}$ emissions, respectively, and the contributions to NMVOC and $NH_3$ emissions are negligible (< 1%). For $SO_2$ and $NO_x$, power

plant is an important emission sources in the BTH region, especially in Tianjin, Shijiazhuang, Tangshan, and Handan.

The emissions from industrial boiler account for 27%, 19%, 8%, 1%, and < 1% of the total $SO_2$, $NO_x$, $PM_{2.5}$, NMVOCs, and $NH_3$ emissions, respectively. As shown in **Fig. 3**, there are many industrial boilers in the BTH region. Industrial boiler is one of the most important emission sources for $SO_2$ and $NO_x$.

The emissions from cement contribute 6%, 9%, and 10% of the total $SO_2$, $NO_x$, and $PM_{2.5}$ emissions, respectively, and the contributions to NMVOC and $NH_3$ emissions are negligible (< 1%). Most of cement plants are located in South and East of Hebei.

The emissions from steel represent 8%, 3%, and 22% of the total $SO_2$, $NO_x$, and $PM_{2.5}$ emissions, respectively, and the contributions to NMVOC and $NH_3$ emissions are negligible (< 1%). Tangshan has the largest number of steel plants in the BTH region, steel accounts for over half of $PM_{2.5}$ emissions in Tangshan.

Besides the aforementioned sectors, 8%, 8%, 13%, 36%, and < 1% of the total $SO_2$, $NO_x$, $PM_{2.5}$, NMVOCs, and $NH_3$ emissions come from other industrial processes (chemistry, coking, nonferrous metal, brick, ceramics, lime, glass, refinery), respectively. Industrial process is the most important emission source for NMVOCs, accounting for nearly half of the emissions in Tianjin and Shijiazhuang.

In total, in the BTH region, industrial sectors (power plant, industrial boiler, cement, steel, and other industrial process) contribute 61%, 55%, 62%, 56%, 58%, 22%, 36% and 0% of the total $SO_2$, $NO_x$, $PM_{10}$, $PM_{2.5}$, BC, OC, NMVOCs, and $NH_3$ emissions in 2014.

Considering the large contribution of industrial sources to total emissions, the application of unit-based method results in remarkable changes in the spatial distribution of air pollutant emissions. The emission rates of $PM_{2.5}$, $NO_x$ and $SO_2$ of the proxy-based and unit-based inventories and their differences are shown in **Fig. 4**. In the unit-based emission inventory, the emission is lower than that in the proxy-based emission inventory in the urban centers of BTH region. Instead, a large amount of the emission is concentrated in certain points in suburban areas, where large plants are located.

## 3.2 Evaluation of the unit-based emission inventory

In order to study the accuracy of the unit-based inventory, the simulation results of $SO_2$, $NO_2$, ozone and $PM_{2.5}$ with the unit-based inventory are compared with the observational data from China National Environmental Monitoring Centre. The observations are available for eighty sites located in 13 cities in the BTH region, including 70 sites in urban area and 10 sites in suburban area. The accurate location of urban and suburban sites in Beijing is shown in Fig.S5-S6. The analysis of the results is shown in **Table 1**. We use normalized mean bias (NMB), normalized mean error (NME), mean fractional bias (MFB) and mean fractional error (MFE) (EPA, 2007) to quantitatively evaluate the model performance.

$SO_2$ and $NO_2$ are precursors of $PM_{2.5}$, so we first compare the simulation results of gaseous pollutants with observations. For $NO_2$, the results with proxy-based inventory overestimates the observations by 22% while results with unit-based inventory overestimates by 9% in January. Similarly, in July, the simulated $NO_2$ concentrations show overestimation in simulations with both inventories but the overestimation is less with unit-based inventory. The simulation results of $SO_2$ is similar to those of $NO_2$. However, the overestimation is higher with both inventories and the differences between the concentrations with two inventories are larger. The overestimation of $SO_2$ concentrations may be due to the lack of several $SO_2$ reaction mechanisms in CMAQ, such as heterogeneous reactions of $SO_2$ on the surface of dust particles (Fu et al., 2016), the oxidation of $SO_2$ by NOx in aerosol liquid water (Cheng et al., 2016;Wang et al., 2016a), the effects of $SO_2$ and $NH_3$ on secondary organic aerosol formation (Chu et al., 2016), etc. It may also be due to uncertainty in emission inventory, especially the uncertainty in the removal efficiencies of $SO_2$ control facilities. The biased spatial distribution of $SO_2$ emissions from residential combustion may also contribute to the overestimation. A large fraction of residential combustion takes place in the rural areas. In this work, however, the emission of residential combustion is allocated by GDP and population, which leads to an overestimation of $SO_2$ emission in urban area and hence an overestimation of $SO_2$ concentration.

For ozone, the simulation results in January with proxy-based inventory underestimate the observations by 21% while the results with unit-based inventory underestimate by only 5%. The simulation results in

July follows the same trend. China is experiencing more and more severe ozone pollution these years (Li et al., 2019), which usually occurs in summer. Therefore, we analyse two extra indices of ozone, 1-hour-peak ozone and daily maximum 8-h averaged (MDA8) ozone concentration in July, which are shown in **Table 2**. The results of 1-hour-peak ozone and MDA8 ozone concentration is similar to that of monthly average ozone concentration. The concentration with the unit-based inventory is slightly higher than that with proxy-based inventory and closer to the observation. The reason for the changes in ozone concentrations will be discussed later.

The simulated $PM_{2.5}$ concentrations with unit-based inventory are lower than that with proxy-based inventory in both winter and summer. In January, the simulated $PM_{2.5}$ concentrations with proxy-based inventory overestimates the observed values by 25% while the overestimation is 7% with unit-based inventory. In July, the simulated $PM_{2.5}$ concentrations with both inventories are 17% and 30% lower than the observations, respectively. An overall underestimation is as expected because the default CMAQ model underestimates the concentrations of secondary organic aerosol (Zhao et al., 2016) significantly and the fugitive dust emission is not included in the emission inventory. According to Boylan and Russell (2006), the simulation results of PM is acceptable when Mean Fractional Bias (MFB) is less than or equal to ±60% and Mean Fractional Bias (MFE) is less than 75% and a model performance goal is met when MFB is less than ±30% and MFE is less than 50%. The statistical indices of the simulation results of $PM_{2.5}$ with both inventories and both months are within the performance goal value, which means that the simulation results are relatively accurate.

**Fig. 5** further shows the spatial distribution of $SO_2$, $NO_2$, ozone, 1-hour-peak ozone, MDA8 ozone and $PM_{2.5}$ concentrations with the proxy-based inventory, the differences between the other two simulations and proxy-based inventory. For $SO_2$, $NO_2$ and $PM_{2.5}$, the concentrations in the urban area is generally higher with proxy-based inventory than that with unit-based inventory, especially in winter. In January, large difference of concentrations of simulations with two inventories are found in urban Tianjin, Tangshan, Baoding and Shijiazhuang, where a large amount of industrial emissions are allocated in the proxy-based inventory due to large population density. The simulation of July follows the same pattern but the concentrations and the difference between the concentrations with two inventories are lower than

those of January. In some areas where many factories are located, such like the northern part of Xingtai city, the concentration with unit-based inventory is higher because of the high emission intensity. There are two reasons for the difference between results with two inventories. The first one is the spatial distribution. With detailed information of industrial sectors, more emissions are allocated to certain

locations in suburban/rural areas in the unit-based emission inventory. From "Diff1" (hypo unit-based minus proxy-based), we can see that the improved horizontal distribution of the unit-based emission inventory significantly decreases the $PM_{2.5}$, $SO_2$, and $NO_2$ concentrations in most urban centers, and significantly increases the concentrations in a large fraction of suburban and rural areas, especially the areas where large industrial plants are located in. The other reason is vertical distribution. Plume rise is

calculated in the simulation with the unit-based inventory, which causes the difference of emissions in vertical layers. The higher the pollutants are emitted, the lower the ground concentration becomes. From the differences between Diff1 and Diff2 we can see that the plume rise leads to lower concentrations over the whole region.

For ozone, the difference of concentration is evident but opposite to that of $PM_{2.5}$. This is because that

urban centers of Beijing/Tianjin are located in the VOC-control chemical regime (Liu et al., 2010). The emissions of $NO_x$ in surface layer are less in the unit-based inventory than in the proxy-based inventory, which leads to higher ozone concentration in urban area.

The spatial distribution of concentrations of these pollutants are significantly heterogeneous. The NME and MFE of most pollutants in two months are lower with unit-based inventory than with proxy-based

inventory, which means the spatial distribution with unit-based inventory agrees more with the observation than that of unit-based inventory. For $SO_2$, $NO_2$ and $PM_{2.5}$, peak concentrations usually occur in the urban center while it's the opposite for ozone. We apply the metric of "concentration gradient", which is defined as the ratios of urban monthly mean concentrations to suburban concentrations, to quantitatively characterize the heterogeneous spatial distributions. We calculate the concentration

gradients for Beijing and Tianjin (**Fig. 6**), since there are both urban and suburb observational sites in these two cities. The concentration gradient of $NO_2$ and $SO_2$ between urban and suburban areas is closer to the observations in the simulation with unit-based inventory than that with proxy-based inventory (**Fig.**

**6**). The simulated $O_3$ concentration gradients with unit-based, proxy-based inventories and the observation are 0.7, 0.5 and 0.9 in January and 0.9, 0.8 and 1.1 in July. As for 1h-peak and MDA8 ozone in July, the simulated results with unit-based inventory is also closer to the observation. As stated previously, this is explained by the VOC-limited photochemical regime and lower $NO_x$ emissions in the unit-based inventory over the urban areas. As for $PM_{2.5}$, the concentration gradients for simulations with unit-based, proxy-based inventories and observations in Beijing are 1.6, 2.1 and 1.5 in January and 1.3, 1.5 and 1.3 in July. The results imply that the unit-based emission inventory better reproduces the distributions of pollutant emissions between the urban and suburban areas.

To further elucidate the reasons for the difference between the $PM_{2.5}$ concentrations with two emission inventories, we examine the simulation results of different chemical components, including sulfate ($SO_4^{2-}$), nitrate ($NO_3^-$), ammonium ($NH_4^+$), element carbon (EC) and organic carbon (OC), as shown in **Fig. 7** and **Table 2**. The concentrations of EC and OC in the simulation with unit-based inventory are generally lower than that with proxy-based inventory in both January and July, especially in urban Beijing, Baoding and Shijiazhuang. This pattern is similar to that of $PM_{2.5}$. In some cities such as Xingtai, the concentrations of EC and OC in the simulation with unit-based inventory are slightly higher than that with proxy-based inventory.

The results of secondary inorganic aerosols are quite different. From **Fig. 7** and **Table 3** we can see that the sulfate concentrations is lower in most areas in the simulation with unit-based inventory as compared to that with proxy-based inventory, which is because that the sensitivity of sulfate concentrations to $SO_2$ concentration is positive during all months (Zhao et al., 2017b). The differences of the concentration of sulfate is similar to that of $SO_2$, which is shown in **Fig. 5**. The difference of ammonium concentration is relatively small compared with other components. As for nitrate, concentration of nitrate in the simulation with unit-based inventory is much higher than that with proxy-based inventory in winter while the differences between the results with two inventories vary with location in summer. Sulfate concentrations in the unit-based approach are much lower than the proxy-based approach whereas ammonium is almost constant as shown in **Fig. 7**. In this case, more $HNO_3$ is converted to $NO_3^-$ with excess $NH_4^+$ whereas these processes depend on abundance of $HNO_3$ or $NH_3$. Taking all chemical components into account, the

primary components account for most of the differences in PM$_{2.5}$ concentrations between the simulations with two inventories. In contrast, however, the complex responses of various secondary components often counteract each other (especially in January), leading to an overall smaller contribution of secondary components to the PM$_{2.5}$ concentration differences.

## 4 Conclusion

In this study, we developed a high-resolution emission inventory of major pollutants for BTH region for year 2014 with unit-based emissions from industrial sectors. The emissions of SO$_2$, NO$_x$, PM$_{10}$, PM$_{2.5}$, BC, OC and NMVOCs from industrial sectors are 869 kt, 1164 kt, 910 kt, 622 kt, 71 kt, 63 kt and 1390 kt respectively, accounting for 61%, 55%, 62%, 56%, 58%, 22% and 36% of the total emissions. The emissions in unit-based emission inventory are lower than that in the proxy-based emission inventory in most urban centers of the BTH region because of the concentrated emissions in point sources. The application of the unit-based emission inventory improves model-observation agreement for most pollutants. The accurate location of point sources leads to lower concentration of primary pollutants in urban area and higher in suburban area. The plume rise accounts for the lower concentration of the whole region. For SO$_2$, NO$_2$ and PM$_{2.5}$, the concentrations in the urban area decrease significantly and become closer to the observations mostly due to the decrease of urban emissions. For ozone, the concentrations in the urban area increase slightly and also show better agreement with observations mainly due to the more reasonable allocation of NO$_x$ emissions. The improvement is particularly significant for the urban-suburban concentration gradients. For PM$_{2.5}$, the concentration gradients for the simulations with unit-based, proxy-based inventories and observations in Beijing are 1.6, 2.1 and 1.5 in January and 1.3, 1.5 and 1.3 in July. For ozone, the corresponding values are 0.7, 0.5 and 0.9 in January and 0.9, 0.8 and 1.1 in July, implying that the unit-based emission inventory better reproduces the distributions of pollutant emissions between the urban and suburban areas.

The unit-based industrial emission inventory enables more accurate source apportionment and more reliable research on air pollution formation mechanism, and therefore contributes to the development of

more precisely targeted control policies. To further improve the emission inventory, it is necessary to improve the spatial allocation of emissions from non-industrial sectors, such as the residential and commercial sectors. Our previous study provides an example to develop a village-based residential emission inventory in rural Beijing (Cai et al., 2018). Such studies on high-resolution emission inventories,

for both industrial and nonindustrial sources, are highly needed and should be extended to other provinces and/or regions as well. In addition, plume-in-grid might help to further improve the model performance, which merits further in-depth study.

## Author contribution

S.W. and B.Z. designed the research. H.Z., S.C. and X.C. performed the research. H.Z., S.C., B.Z., S.W.

and X.C. analyzed the results. H.Z., S.C., B.Z., S.W., X.C. and J.H. wrote the paper.

## Acknowledgements

This research has been supported by the National Natural Science Foundation of China (21625701), Strategic Priority Research Program of Chinese Academy of Sciences (XDA20040502), the Ministry of Environmental Protection of China (DQGG0301) and Beijing Municipal Commission of Science and

Technology (D171100001517001). The simulations were completed on the "Explorer 100" cluster system of Tsinghua National Laboratory for Information Science and Technology.

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

## Figures

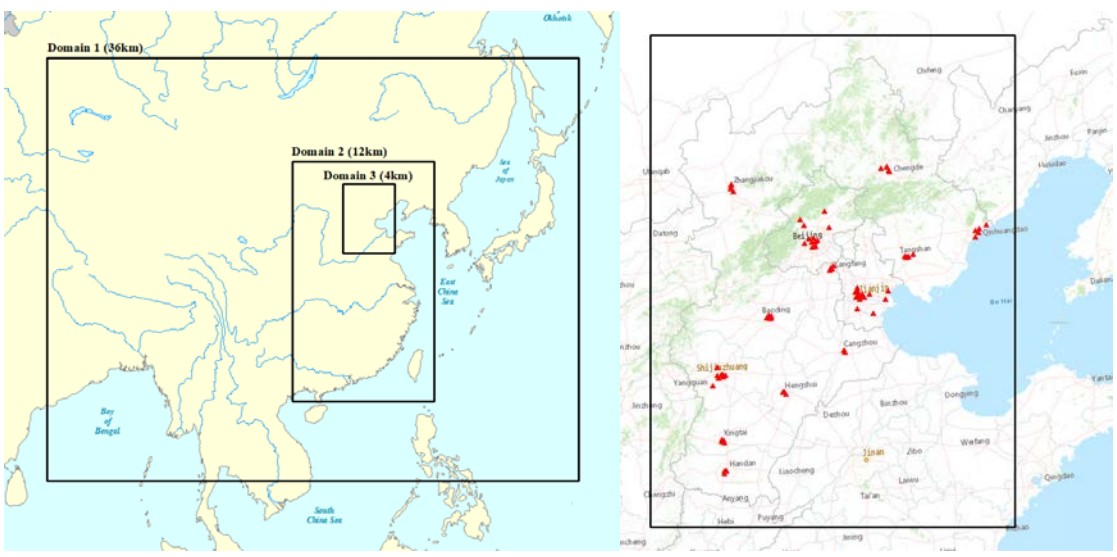

**Fig. 1 The three-domain nested CMAQ domain (left) and the observational sites in Beijing-Tianjin-Hebei region (right)**

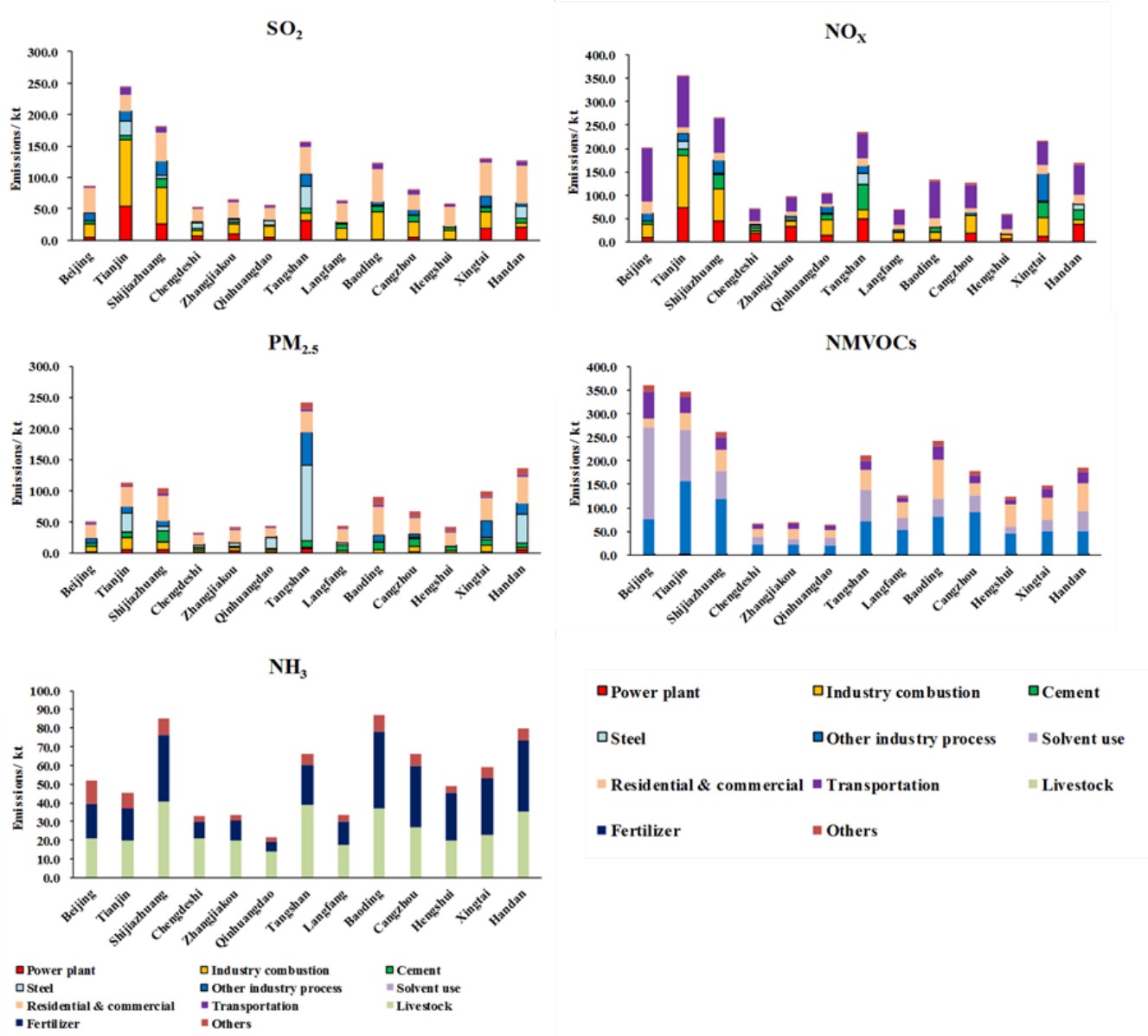

**Fig. 2 Sectoral contributions to emissions in BTH region in 2014**

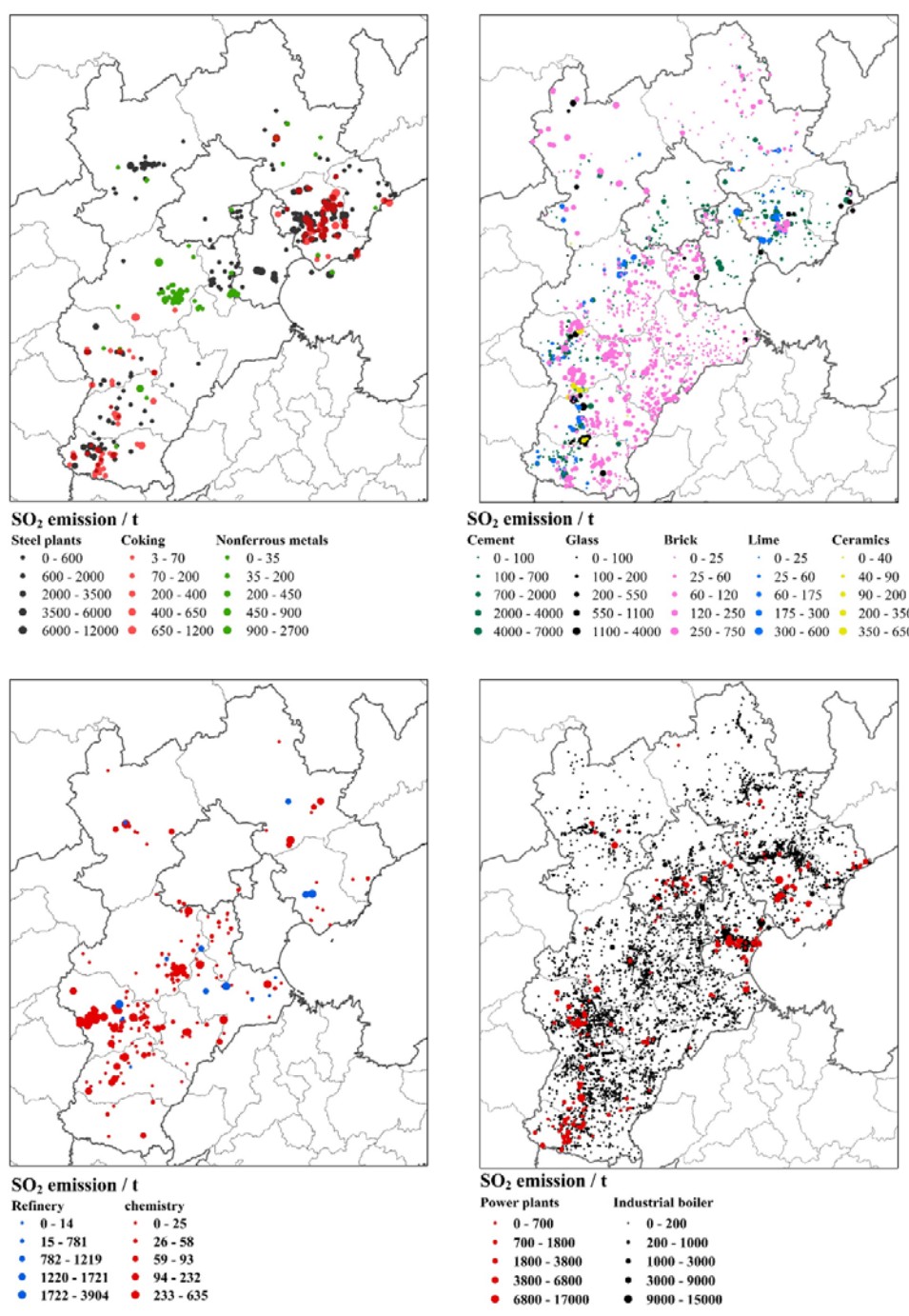

**Fig. 3 Locations and emissions of industrial sources in the BTH region. The industrial plants are divided into four groups to display more clearly.**

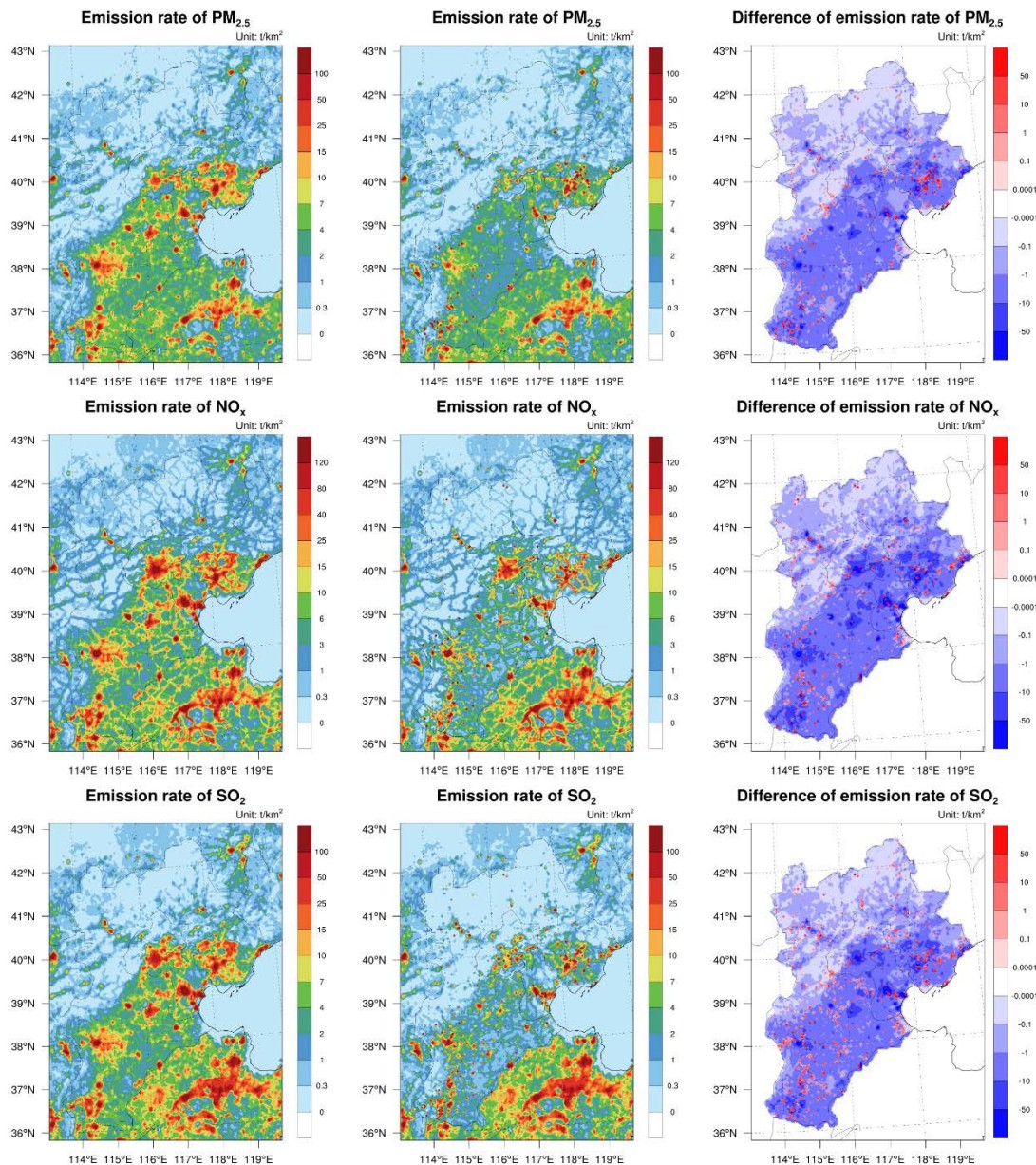

**Fig. 4 Emission rate of PM2.5, NOx and SO2 emissions of the proxy-based (left column) and unit-based (middle column) inventories and their differences (unit-based minus proxy-based, right column). Note that the emissions are the same in provinces other than Beijing, Tianjin, and Hebei.**

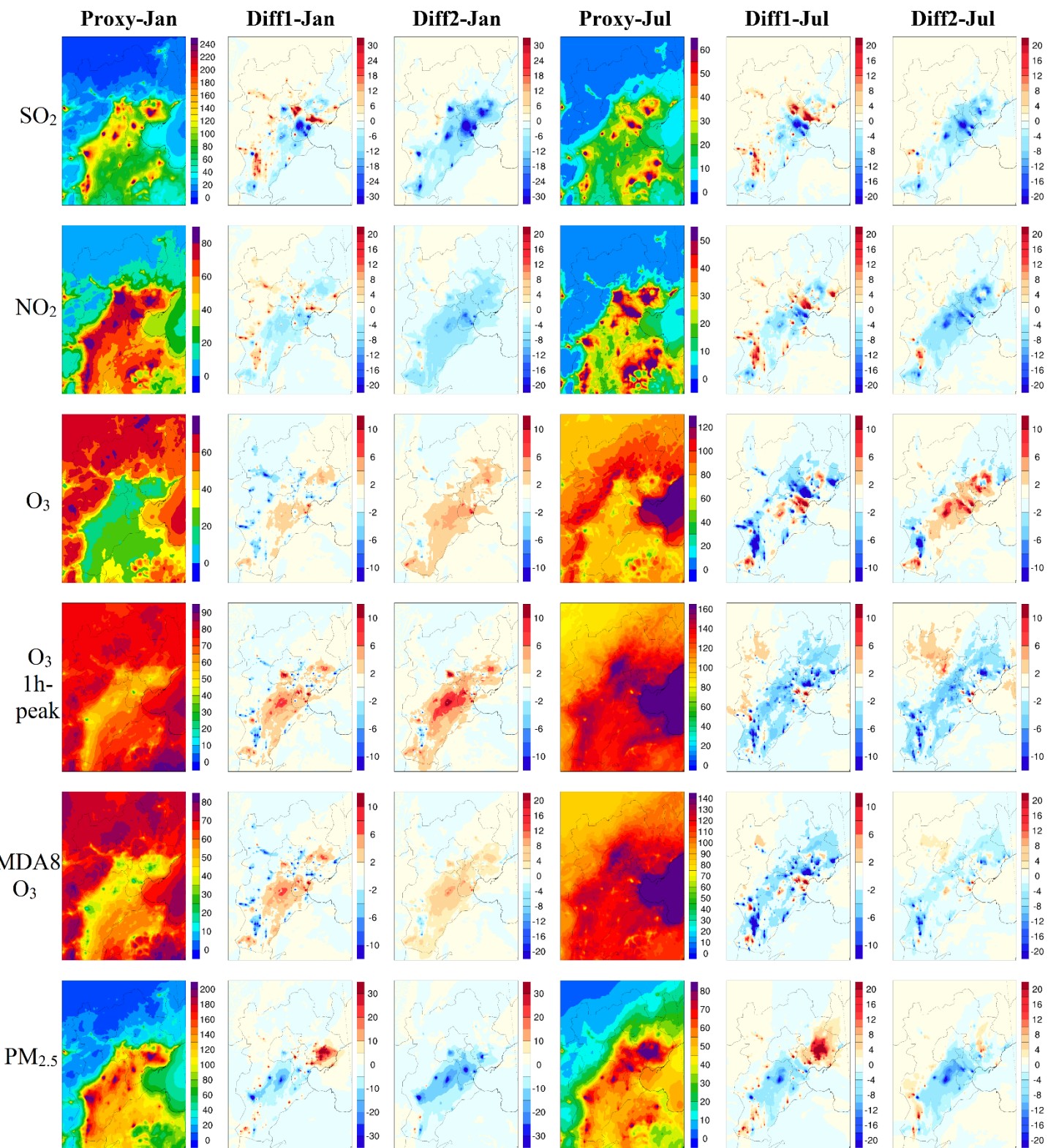

**Fig. 5 Spatial distribution of the monthly (January and July) mean concentrations of SO₂, NO₂, ozone, 1h-peak ozone, MDA8 ozone and PM₂.₅ with the proxy-based inventory, and the differences between the other two simulations and proxy-based inventory (Diff1: hypo unit-based minus proxy-based; Diff2: unit-based minus proxy-based). The units are μg/$m^3$ for all panels.**

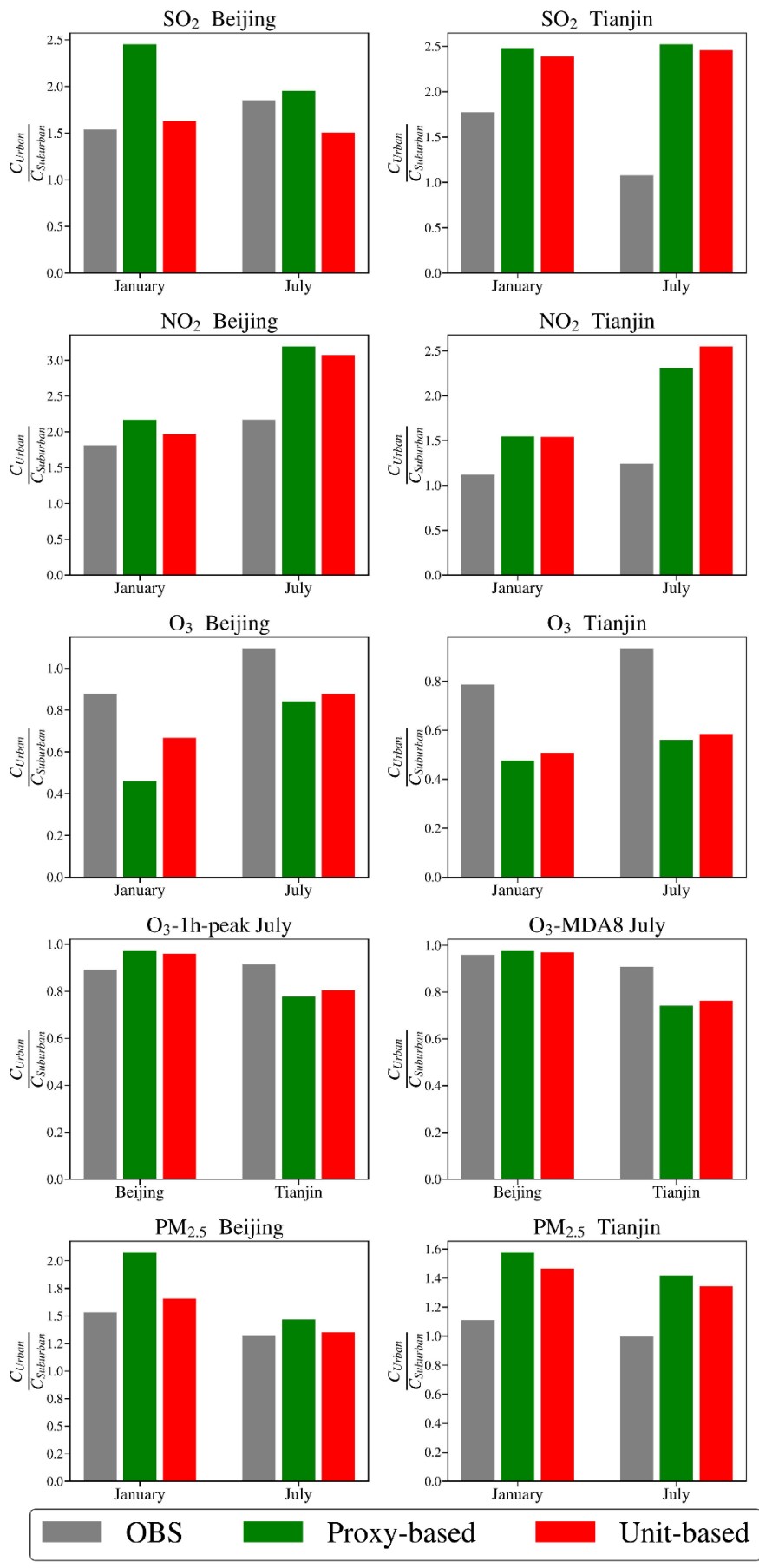

**Fig. 6 Observed and simulated concentration gradients of NO₂, PM₂.₅, ozone and SO₂ with the proxy-based and unit-based inventories in Beijing (left) and Tianjin (right)**

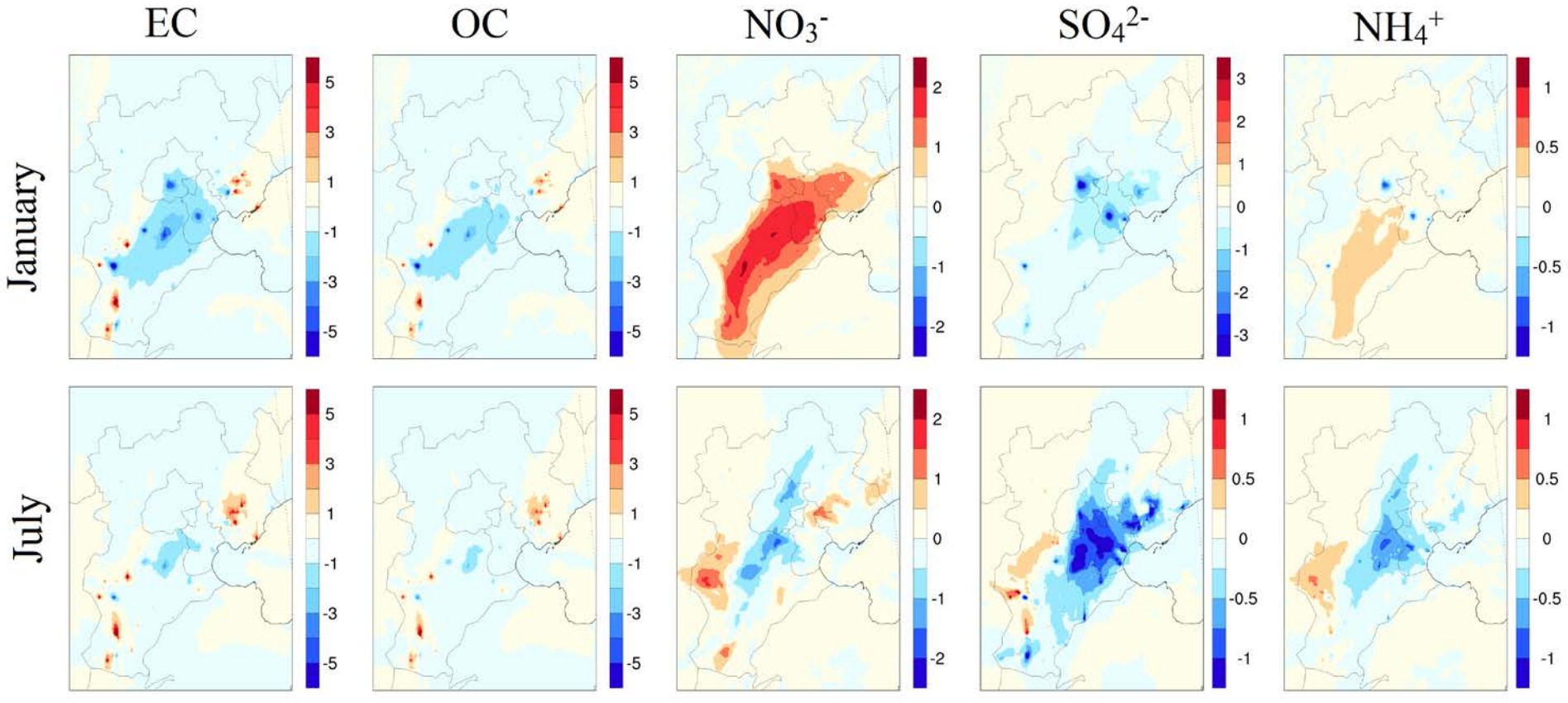

**Fig. 7 The differences (unit: μg/$m^3$) in the simulation results of the components of PM$_{2.5}$ between the results with two inventories (unit-based minus proxy-based).**

**Table 1. The statistics for model performance of $PM_{2.5}$, $NO_2$, $SO_2$, 1-hour-peak ozone and daily maximum 8-h averaged (MDA8) ozone in January and July of 2014 with proxy-based and unit-based inventories**

| Month | Species | Emission | SIM ($\mu g/m^3$) | OBS ($\mu g/m^3$) | NME | NMB | MFB | MFE |
|---|---|---|---|---|---|---|---|---|
| Jan | $SO_2$ | Proxy-based | 251.9 | 112.3 | 131% | 124% | 51% | 57% |
| | | Unit-based | 207.8 | | 93% | 85% | 35% | 42% |
| | $NO_2$ | Proxy-based | 88.0 | 72.0 | 30% | 22% | 14% | 19% |
| | | Unit-based | 77.9 | | 23% | 8% | 5% | 16% |
| | $O_3$ | Proxy-based | 16.8 | 21.4 | 36% | -21% | -19% | 27% |
| | | Unit-based | 20.2 | | 33% | -6% | -6% | 22% |
| | $PM_{2.5}$ | Proxy-based | 176.3 | 141.1 | 39% | 25% | 12% | 22% |
| | | Unit-based | 151.5 | | 31% | 7% | 2% | 20% |
| Jul | $SO_2$ | Proxy-based | 58.4 | 26.4 | 140% | 121% | 54% | 63% |
| | | Unit-based | 42.7 | | 86% | 62% | 34% | 47% |
| | $NO_2$ | Proxy-based | 61.5 | 35.9 | 80% | 72% | 33% | 40% |
| | | Unit-based | 52.1 | | 62% | 45% | 20% | 34% |
| | $O_3$ | Proxy-based | 64.0 | 66.8 | 96% | -4% | -26% | 26% |
| | | Unit-based | 69.0 | | 90% | 3% | -21% | 22% |
| | $PM_{2.5}$ | Proxy-based | 71.2 | 85.5 | 26% | -17% | -12% | 19% |
| | | Unit-based | 60.1 | | 34% | -30% | -21% | 25% |
| Two-month average | $SO_2$ | Proxy-based | 155.2 | 69.4 | 133% | 124% | 53% | 60% |
| | | Unit-based | 125.2 | | 92% | 81% | 35% | 45% |
| | $NO_2$ | Proxy-based | 74.7 | 53.9 | 47% | 39% | 23% | 30% |
| | | Unit-based | 65.0 | | 36% | 21% | 13% | 25% |
| | $O_3$ | Proxy-based | 40.4 | 44.1 | 82% | -8% | -22% | 27% |
| | | Unit-based | 44.6 | | 76% | 1% | -14% | 22% |
| | $PM_{2.5}$ | Proxy-based | 123.8 | 113.3 | 34% | 9% | 0% | 21% |
| | | Unit-based | 105.8 | | 32% | -7% | -10% | 23% |

**Table 2 The statistics for model performance of 1-hour-peak ozone and daily maximum 8-h averaged (MDA8) ozone concentration in July of 2014 with proxy-based and unit-based inventories**

| Species | Emission | SIM μg/$m^3$ | OBS μg/$m^3$ | NME | NMB | MFB | MFE |
|---------|----------|------|------|-----|-----|-----|-----|
| 1h-peak ozone | Proxy-based | 133.7 | 171.2 | 28% | -22% | -22% | 32% |
| | Unit-based | 135.0 | | 27% | -21% | -21% | 31% |
| MDA8 ozone | Proxy-based | 115.1 | 128.1 | 23% | -10% | -9% | 25% |
| | Unit-based | 117.1 | | 22% | -9% | -7% | 24% |

**Table 3. The mean concentrations (unit: μg/$m^3$) of the components of PM2.5 with proxy-based and unit-based inventories and their differences**

| Month | Emission | EC | OC | $NO_3^-$ | $SO_4^{2-}$ | $NH_4^+$ |
|-------|----------|-----|-----|-----|-----|-----|
| Jan | Proxy-based | 41.2 | 49.7 | 11.8 | 11.7 | 7.8 |
| | Unit-based | 38.5 | 48.0 | 13.0 | 10.2 | 7.6 |
| | difference | -7% | -4% | 10% | -12% | -2% |
| Jul | Proxy-based | 8.3 | 9.3 | 11.9 | 10.2 | 7.3 |
| | Unit-based | 7.1 | 8.4 | 11.8 | 9.3 | 6.9 |
| | difference | -15% | -9% | 0% | -9% | -5% |