# Peer review of "Development of a unit-based industrial emission inventory in the Beijing-Tianjin-Hebei region and resulting improvement in air quality modelling"

_Atmospheric Chemistry and Physics, 2018_

## Referee Comment (RC1) · Anonymous Referee #1 · 5 Nov 2018

This study developed the unit-based industrial emission inventory in Beijing-Tianjin-Hebei region, for which configurations and locations of individual industrial emission sources were utilized. Significant differences in horizontal distributions of emissions were seen by comparing with the traditional proxy-based emission inventory. The air quality simulations using this unit-based emission inventory showed better model performance than the proxy-based emission inventory.

I think this is an important progress to get better model performance. It should contribute to developing effective emission controls against heavy air pollution in this region. However, various critical information is missing in the current manuscript. It is necessary to revise it based on the comments described below.

As mentioned in the introduction, previous studies have already developed unit-based emission inventories while their target sectors may be limited. I suppose there should be more papers including Liu et al. (2015) for example. It is necessary to clearly describe what is new in this study. This manuscript says previous studies did not cover all industrial sectors in the BTH region. Then, does this study cover all industrial sectors? Which sectors were newly included? Is the methodology identical for the sectors which have been already included in previous studies? Significance of this study should be described more clearly.

One of difficulties in unit-based emission inventories we often face is consistency of energy consumption against energy statistics. Did this study use energy consumption reported from each emission source? If so, is the sum of the reported energy consumption consistent with that in energy statistics? Usually, it is very hard to collect detailed information of small emission sources. If this is the case, energy consumption should not be consistent, and a hybrid approach in which unit-based and proxy-based information are combined may be necessary for each sector. The unit-based and proxy-based emission inventories were compared in this study. Do energy consumptions used in both inventories match?

Although detailed descriptions for vertical distributions are missing in the current manuscript, I agree that reasons of differences in concentrations between the unit-based and proxy-based emission inventories should be horizontal distributions and vertical distributions as mentioned in the second paragraph in the page 9. According to Figures 5 and 7, concentrations simulated with the proxy-based emissions are almost entirely lower throughout the domain. If influences of horizontal distributions are dominant, it is supposed that concentrations in surrounding regions would become higher, but such influences seem to be very limited. Therefore, it might be possible that differences in concentrations between two emission inventories are mainly caused by differences in vertical distributions of emissions. I would strongly recommend conducting an additional simulation to separate influences of horizontal and vertical distributions of emissions by changing only each of them.

This paper shows relative improvements in the unit-based emission inventory by comparing with the proxy-based emission inventory. Therefore, relative changes depend not only on the unit-based inventory but also the proxy-based inventory. If poor proxies are used in the proxy-based inventory, relative improvements could become larger. Therefore, it is important to explicitly shows which proxies were used in the proxy-based inventory for each sector (not just "such as population ..." at the end of the section 2.2). Use of better proxies should be also one of possible directions to get better model performance.

Specific comments are as follows.

Page 3, Line 9-10

I think that Lim et al. (2005) is not related to the description around here.

Page 3, Line 17-18

It is not clear which sectors are considered in previous studies and which sectors newly appear in this study. I would recommend adding a table listing all the industrial sectors considered and which are new in this study.

Page 4, Line 6-7

It is not clear what kind of product yields are used for estimating emissions of each sector. I would recommend showing types of products used for each sector in a table I recommended above.

Page 4, Lines 9 and 17

The equation (1) is used to estimate emissions of the pollutant i. The industrial enterprise j and the production process m appear in this equation, but they are summed up.

Then, how about the control technology n? It is not summed up, but it does not appear in the left-hand side. Usually fractions of control technologies are inserted, then they are summed up for all of control technologies. This is the same for the control technology k in the equation (2).

Page 4, Lines 9 and 17

I do not understand why the equations (1) and (2) are separated. It seems the first and second terms of the equation (2) represent clinker and cement production, respectively. However, isn't it possible to treat both as one of production processes m? If not, then what are production processes considered in both equations? Please clarify them. In fact, it is not clear what production processes considered in this study are.

Page 4, Lines 12-14

EFs depend only on the pollutant i and the production process m. Is there any possibility to use emission factors specific to each industrial enterprise? Is it enough to use identical emission factors for all the industrial enterprises?

Page 4, Line 25 – Page 5, Line 1

Specific references are not listed here while a lot of specific references for proxy-based emissions are listed in a subsequent paragraph. Specific references should be also listed for unit-based emissions as much as possible.

Page 5, Lines 2-4

Do these numbers cover all the plants located in the target area?

Page 5, Line 5

Is there no information on control technologies for boilers?

Page 5, Line 9

Is the expression "emission factor method" appropriate? The unit-based approach also

uses emission factors. I think it is usually called as "top-down method" (but sometimes confused with top-down estimates utilizing observations including satellites).

Page 5, Line 17

How about speciation of PM2.5 and NMVOCs for unit-based emissions?

Page 5, Line 20 – Page 6, Line 18

References for models and modules are required.

Page 6, Line 10

What are "other" configurations? Please show explicitly.

Page 6, Lines 21-23

Is CO not included in this study? Why?

Page 7 Lines 1-22

Area names are mentioned in these paragraphs. However, horizontal distributions firstly appear later in Fig. 3. Its description should appear before descriptions of areas.

Page 7, Line 6

It is impossible to see many industrial boilers in Fig. 2.

Page 8, Line 9

I think that NMB and NME are not appropriate metrics in terms of this study. The target of this study is accurate horizontal distributions. However, overestimation in one areas and underestimation in other areas could be cancelled out in these metrics. It is necessary to appropriate metrics which can properly shows improvements realized in this study.

Page 8, Lines 15-17

What is a possible reason for the poor model performance on SO2?

Page 9, Lines 19-20

I cannot find any descriptions on plume rise before here. How to gather stack information? How to calculate plume rise? These descriptions are required in the method section.

Page 10, Line 1

Details of "concentration gradient" are necessary. How to select urban and suburban locations? Are monthly mean concentrations used?

Page 10, Lines 24-27

I think it is not enough to explain changes of $NO_3^-$ only by $NO_x$ sensitivities. I do not think they are main reasons. $SO_4^{2-}$ concentrations in the unit-based approach are much lower than the proxy-based approach whereas $NH_4^+$ is almost constant as shown in Fig. 7. In this case, more $HNO_3$ is converted to $NO_3^-$ with excess $NH_4^+$ whereas these processes depend on abundance of $HNO_3$ or $NH_3$.

Reference

Liu, F., Zhang, Q., Tong, D., Zheng, B., Li, M., Huo, H., and He, K. B.: High-resolution inventory of technologies, activities, and emissions of coal-fired power plants in China from 1990 to 2010, Atmos. Chem. Phys., 15, 13299-13317, https://doi.org/10.5194/acp-15-13299-2015, 2015.

---

## Referee Comment (RC2) · Anonymous Referee #2 · 10 Dec 2018

This is a timely paper that describes the development of a unit-based industrial emission inventory in northern China, which still suffers severe air pollution even though the government has put tremendous amount of effort in emission controls. A detailed, united-based emission inventory will be of great value when air quality models are used in developing/assessing emission control strategies. The paper is generally well-written. I would recommend the paper be published in ACP after addressing my comments below:

1. The paper lacks details on how vertical distribution of point source emissions are

[Figure]

treated in the simulation. In the results section, it is mentioned that plume rise contributes to the difference between the CMAQ results. However, no details were provided on how the parameters needed for plume rise calculations are obtained. In my understanding, such data are not universally available (even in the US) so presumably the same situation is applicable in China. What is the criteria for selecting point sources for plume rise calculation and how missing information is estimated. I also believe that the authors should perform off-line emission vertical distribution calculations and compare with the empirical vertical distribution used for the proxy-based emission inventory. For many of people without access to the detailed unit-based emission inventory, it will be useful to see this information so that vertical distribution in the traditional inventories can also be improved.

2. One of the major conclusions from the study is that unit-based emission inventory leads to significant improvement in the model performance. However, the only quantitative assessment is monthly average concentrations of SO2, NO2, O3, PM2.5 using all the stations in the domain. This is not sufficient as information is lost in the averaging process. At minimal, the authors should show performance of these pollutants at each individual sites. Time series should also be shown for sites with significant differences. It will help identify the cause of the differences. For O3, it is necessary to show performance of 1-hr peak ozone and 8-hr daily maximum. Very large error still exists for SO2. More discussion of this over-estimation should be included.

3. Table 1 shows "annual average" but only January and July simulations were performed. How did you calculate annual average with only two months of simulation?

---

## Author Comment (AC1) · 28 Jan 2019

Reviewer 1:

This study developed the unit-based industrial emission inventory in Beijing-Tianjin-Hebei region, for which configurations and locations of individual industrial emission sources were utilized. Significant differences in horizontal distributions of emissions were seen by comparing with the traditional proxy-based emission inventory. The air quality simulations using this unit-based emission inventory showed better model performance than the proxy-based emission inventory.

I think this is an important progress to get better model performance. It should contribute to developing effective emission controls against heavy air pollution in this region. However, various critical information is missing in the current manuscript. It is necessary to revise it based on the comments described below.

Response: We appreciate the reviewer's valuable comments which help us improve the quality of the manuscript. We have carefully revised the manuscript according to the reviewers' comments. Point-to-point responses are given below. The original comments are in black, while our responses are in blue.

(1) As mentioned in the introduction, previous studies have already developed unit-based emission inventories while their target sectors may be limited. I suppose there should be more papers including Liu et al. (2015) for example. It is necessary to clearly describe what is new in this study. This manuscript says previous studies did not cover all industrial sectors in the BTH region. Then, does this study cover all industrial sectors? Which sectors were newly included? Is the methodology identical for the sectors which have been already included in previous studies? Significance of this study should be described more clearly.

Response: We thank the reviewer for this valuable comment. We searched the papers about unit-based emission inventories again and added more papers in the Introduction section, including Liu et al. (2015) about emission from coal-fired power plants, Chen et al. (2015) about emission from cement industry and Wu et al. (2015) about emission from steel industry. (Page 3, Line 13-16) In the previous studies, they usually focus on one or several sectors such as power plant, cement plant, and iron plant. In this study, we cover most industrial sectors including power plant, industrial boiler, iron and steel production, non-ferrous metal smelter, coking, cement, glass, brick, lime, ceramics, refinery, and chemical industries (Page 4, Line 3-5). Compared with most previous studies, industrial boiler, non-ferrous metal smelter, coking, glass, brick, lime, ceramics, refinery, and chemical industries are newly included. The methodology of calculating the emission of point sources is similar to previous studies, but we calculate the emissions from cement and iron sectors according to specific industrial processes, such as clinker burning and clinker processing stages in the cement sector (Page 4, Line 18 to Page 5, Line 5).

(2) One of difficulties in unit-based emission inventories we often face is consistency of energy consumption against energy statistics. Did this study use energy consumption reported from each emission source? If so, is the sum of the reported energy consumption consistent with that in energy statistics? Usually, it is very hard to collect detailed information of small emission sources. If this is the case, energy consumption should not be consistent, and a hybrid approach in which unit-based and proxy-based information are combined may be necessary for each sector. The unit-based and proxy-based emission inventories were compared in this study. Do energy consumptions used in both inventories match?

Response: Yes, this study calculated emissions using energy consumption or industrial production reported for each emission source.

The plants in this study are from compilation of power industry statistics (China Electricity Council, 2015), China Iron and Steel Industry Association (http://www.chinaisa.org.cn), China Cement Association (http://www.chinacca.org), Chinese environmental statistics (collected from provincial environmental protection bureaus), the first national census of pollution sources (National Bureau of Statistics (NBS), 2010) and bulletin of desulfurization and denitrification facilities from Ministry of Ecology and Environment of China (http://www.mee.gov.cn). (Page 5, Line 6-13)

We compared the sum of the energy consumption or industrial production for each plant with those in official statistics. The sum of individual plants generally accounts for over 90% of the energy consumption or product yield reported in the statistics. For the plants not included in the preceding data sources, we calculate the emission by using "top-down method" and allocate the emission with proxies, such as GDP and population. Therefore, the total energy consumption of both inventories match. (Page 6, Line 14-17; Page 7, Line 25-26)

(3) Although detailed descriptions for vertical distributions are missing in the current manuscript, I agree that reasons of differences in concentrations between the unit-based and proxy-based emission inventories should be horizontal distributions and vertical distributions as mentioned in the second paragraph in the page 9. According to Figures 5 and 7, concentrations simulated with the proxy-based emissions are almost entirely lower throughout the domain. If influences of horizontal distributions are dominant, it is supposed that concentrations in surrounding regions would become higher, but such influences seem to be very limited. Therefore, it might be possible that differences in concentrations between two emission inventories are mainly caused by differences in vertical distributions of emissions. I would strongly recommend conducting an additional simulation to separate influences of horizontal and vertical distributions of emissions by changing only each of them.

Response: We thank the reviewer for this valuable comment. We have conducted an additional simulation in which the unit-based inventory is used but the emission heights are assumed to be the same as the proxy-based inventory. The amount of emission is the same as the other two scenarios. We call the inventory used in this simulation "hypo unit-based inventory".

Fig. R1 (Fig. 5 in the revised manuscript) shows the distribution of the monthly (January and July) mean concentrations of $SO_2$, $NO_2$, ozone, daily maximum 1-h averaged ozone, daily maximum 8-h averaged ozone and $PM_{2.5}$ simulated with the proxy-based inventory, and the differences between the proxy-based simulation and the other two simulations (Diff1: hypo unit-based minus proxy-based; Diff2: unit-based minus proxy-based). For $SO_2$, $NO_2$ and $PM_{2.5}$, the concentrations in the urban area are generally higher with the proxy-based inventory than those with the unit-based inventory, especially in winter. In January, large concentration differences between simulations with two inventories are found in urban Tianjin, Tangshan, Baoding and Shijiazhuang, where a large amount of industrial emissions is allocated in the proxy-based inventory due to large population density. The simulation of July follows the same pattern but the concentrations and the difference between the concentrations with two inventories are lower than those of January. In some areas where many factories are located, such as the northern part of Xingtai city, the concentration with unit-based inventory is higher because of a high emission intensity. There are two reasons for the difference between results with proxy-based and unit-based inventories. The first one is the spatial distribution. With detailed information of industrial sectors, more emissions are allocated to certain locations in suburban/rural areas in the unit-based emission inventory. From "Diff1" (hypo unit-based minus proxy-based), we can see that the improved horizontal distribution of the unit-based emission inventory significantly decreases the $PM_{2.5}$, $SO_2$, and $NO_2$ concentrations in most urban centers, and significantly increases the concentrations in a large fraction of suburban and rural areas, especially the areas where large industrial plants are located in. The other reason is vertical distribution. Plume rise is calculated in the simulation with the unit-based inventory, which causes the difference of emissions in vertical layers. The higher the pollutants are emitted, the lower the ground concentration becomes. From the differences between Diff1 and Diff2 we can see that the plume rise leads to lower concentrations over the whole region.

The results of the additional simulation have been added to the revised manuscript (Page 11, Line 6 to 26; Page 14, Line 2-4)

[Figure]

Fig. R1 Spatial distribution of the monthly (January and July) mean concentrations of SO₂, NO₂, ozone, daily maximum 1-h averaged ozone, daily maximum 8-h averaged ozone and PM₂.₅ simulated with the proxy-based inventory, and the differences between the proxy-based simulation and the other two simulations (Diff1: hypo unit-based minus proxy-based; Diff2: unit-based minus proxy-based). The units are $\mathbf{\mu g/m^3}$ for all panels.

(4) This paper shows relative improvements in the unit-based emission inventory by comparing with the proxy-based emission inventory. Therefore, relative changes depend not only on the unit-based inventory but also the proxy-based inventory. If poor proxies are used in the proxy-based inventory, relative improvements could become larger. Therefore, it is important to explicitly show which proxies were used in the proxy-based inventory for each sector (not just "such as population …" at the end of the section 2.2). Use of better proxies should be also one of possible directions to get better model performance.

Response: For the proxies of each sector, we refer to Zhao et al. (2013), Streets et al. (2003) and Woo et al. (2003). We allocate the emissions of each province and each pollutant by two steps. The first step is to allocate the total emission to each county. The second step is to allocate the emission of each county to each grid. The proxies used in this study are shown in Table R1 (Table S2 in the revised manuscript).

Table R1 Proxies used in the proxy-based inventory for each sector

| Sector | Allocate to county | Allocate to grid |
| --- | --- | --- |
| Power plant, steel, cement | GDP of secondary industry | Population density |
| Industrial combustion, other industrial process | GDP of secondary industry | Population density |
| Domestic fuel | Total GDP | Population density |
| Domestic biomass | GDP of first industry | Population density |
| Transportation | GDP of tertiary industry | Road network |
| Open burning | GDP of first industry | Population density |
| Livestock | GDP of first industry | Population density |
| Fertilizer application | GDP of first industry | Population density |
| Domestic solvent use | Total GDP | Population density |
| Industrial solvent use | GDP of secondary industry | Population density |

(5) Page 3, Line 9-10
I think that Lim et al. (2005) is not related to the description around here.

Response: It is removed from the manuscript.

(6) Page 3, Line 17-18
It is not clear which sectors are considered in previous studies and which sectors newly appear in this study. I would recommend adding a table listing all the industrial sectors considered and which are new in this study.

Response: As is shown in Table R2. The underlined sectors are newly added to this study. This table is added to SI. (Table S3)

Table R2 Comparison of industrial sectors covered in previous studies and this study (the underlined sectors are newly included in this study).

| Study | Sector | Region |
| --- | --- | --- |
| Zhao et al. (2008), Chen et al. (2014), Liu et al. (2015), Li et al. (2017) | Power plants | China |
| Wang et al. (2016b), Wu et al. (2015) | Iron plants | China |

| Lei et al. (2011), Chen et al. (2015) | Cement plants | China |
|---|---|---|
| Qi et al. (2017) | Power plants, iron plants, cement factories, coking factories, heating plants, other industries | BTH |
| This study | Power plants, iron plants, cement factories, coking factories, nonferrous metals, glass factories, brick factories, lime factories, ceramics factories, refinery factories, chemical plants, industrial boilers | BTH |

(7) Page 4, Line 6-7

It is not clear what kind of product yields are used for estimating emissions of each sector. I would recommend showing types of products used for each sector in a table I recommended above.

Response: The types of products used for each sector are listed as follows and in Table S4 of the revised manuscript.

Table R3 Types of products or energy consumption used for estimating emissions of each sector.

| Industrial sector | Product or energy consumption |
|---|---|
| Power plant | Energy consumption |
| Industrial boiler | Energy consumption |
| Iron and steel production | Pig iron, crude steel, rolled steel |
| Non-ferrous metal smelter | Alumina, aluminum, copper |
| Coking | Coke |
| Cement | Cement, clinker |
| Glass | Glass |
| Brick | Brick |
| Lime | Lime |
| Ceramics | Ceramics |
| Refinery | Crude oil, ethylene |
| Chemical industries | Ammonia, caustic soda, soda ash, sulfuric acid, nitric acid |

(8) Page 4, Lines 9 and 17

The equation (1) is used to estimate emissions of the pollutant i. The industrial enterprise j and the production process m appear in this equation, but they are summed up. Then, how about the control technology n? It is not summed up, but it does not

appear in the left-hand side. Usually fractions of control technologies are inserted, then they are summed up for all of control technologies. This is the same for the control technology k in the equation (2).

Response: Equation (1) and equation (2) are revised as follows:

$$E_{i,j} = A_j \times EF_{i,j} \times (1 - \eta_{i,j}) \qquad (1)$$

where $E_{i,j}$ is emissions of pollutant i from industrial enterprise j, $A_j$ is activity level of industrial enterprise j, $EF_{i,j}$ is uncontrolled emission factor of pollutant i from industrial enterprise j, and $\eta_{i,j}$ is removal efficiency of pollutant i by control technology in enterprise j. $\eta_{i,j}$ is determined by the production process and control technology of the industrial enterprise. The $EF_{i,j}$, which depends on the production process of the industrial enterprise, are calculated according to the sulfur and ash contents of fuels (e.g. coal) used in each province (for PM and $SO_2$), or obtained from our previous study (Zhao et al., 2013) (for other pollutants).

For those industrial sources with multiple production processes, such as iron and steel production and cement production, emissions are calculated by using the following equation:

$$E_{i,j} = \sum_m \left( AK_{j,m} \times EF_{i,m} \times (1 - \eta_{i,j,m}) \right) + \left( AC_j \times ef_i \times (1 - \eta_{i,j}) \right) \qquad (2)$$

where $E_{i,j}$ is emissions of pollutant i from industrial enterprise j, $AK_{j,m}$ is the amount of clinker produced by the clinker burning process m of the enterprise j, $EF_{i,m}$ is uncontrolled emission factor for pollutant i from the clinker burning process m, $\eta_{i,j,m}$ is removal efficiency of pollutant i from the clinker burning process m in enterprise j, $AC_j$ is the amount of cement produced by enterprise j, $ef_i$ is uncontrolled emission factors from the clinker processing stage ($ef_i$=0 if i is not particulate matter), $\eta_{i,j}$ is removal efficiency of pollutant i in enterprise j. $\eta_{i,j,m}$ and $\eta_{i,j}$ both depend on the control technology of the industrial enterprise. (Page 4, Line 10 to Page 5, Line 2)

(9) Page 4, Lines 9 and 17
I do not understand why the equations (1) and (2) are separated. It seems the first and second terms of the equation (2) represent clinker and cement production, respectively. However, isn't it possible to treat both as one of production processes m? If not, then what are production processes considered in both equations? Please clarify them. In fact, it is not clear what production processes considered in this study are.

Response: Equations (1) and (2) cannot be merged because the production processes represented by the first and second terms of equation (2) are frequently performed in different enterprises. For example, for cement production, clinker may be produced in one enterprise and subsequently processed in another enterprise, which is very common.

Most industrial sources are calculated by equation (1). Only a few industrial sources with multiple processes, such as steel production and cement production, are calculated by equation (2). We have added the preceding descriptions in the revised manuscript (Page 4, Line 18-19; Page 5, Line 3-5).

(10) Page 4, Lines 12-14
EFs depend only on the pollutant i and the production process m. Is there any possibility to use emission factors specific to each industrial enterprise? Is it enough to use identical emission factors for all the industrial enterprises?
Response: For $SO_2$ and PM, EFs are calculated according to the sulfur and ash contents of fuels (e.g. coal) in each province. For other pollutants, EFs depend only on the pollutant and the production process, and are obtained from our previous studies (Zhao et al., 2013). (Page 4, Line 14-17)
We agree with the reviewer that it is better to use emission factors specific to each individual enterprise. However, such detail offline emission measurements are not yet available in China. The continuous emission monitoring systems (CEMS) data may help to improve the emission esitmates. Cui et al. (2018) estimated the emissions of air pollutants from power plants in China based on the data of CEMS, environmental statistics and the data of pollutant emission permits. (Karplus et al., 2018) evaluated the impact of China's new air pollution standards on $SO_2$ emissions by comparing newly available data from CEMS with satellite measurements. We will work on it in the future.

(11) Page 4, Line 25 – Page 5, Line 1
Specific references are not listed here while a lot of specific references for proxy-based emissions are listed in a subsequent paragraph. Specific references should be also listed for unit-based emissions as much as possible.
Response: The references were added in this manuscript.
For all power and industrial sources except industrial boilers, we collect their detailed information, including latitude/longitude, annual product, production technology/process, and pollution control facilities from compilation of power industry statistics (China Electricity Council, 2015), China Iron and Steel Industry Association (http://www.chinaisa.org.cn), China Cement Association (http://www.chinacca.org), Chinese environmental statistics (collected from provincial environmental protection bureaus), the first national census of pollution sources (National Bureau of Statistics (NBS), 2010) and bulletin of desulfurization and denitrification facilities from Ministry of Ecology and Environment of China (http://www.mee.gov.cn). (Page 5, Line 6-13)

(12) Page 5, Lines 2-4
Do these numbers cover all the plants located in the target area?
Response: It's very difficult to cover all the plants located in Beijing-Tianjin-Hebei region because there are some very small factories. The plants in this study are from compilation of power industry statistics (China Electricity Council, 2015), China Iron and Steel Industry Association (http://www.chinaisa.org.cn), China Cement Association (http://www.chinacca.org), Chinese environmental statistics (collected

from provincial environmental protection bureaus), the first national census of pollution sources (National Bureau of Statistics (NBS), 2010) and bulletin of desulfurization and denitrification facilities from Ministry of Ecology and Environment of China (http://www.mee.gov.cn). The sum of individual plants generally accounts for over 90% of the energy consumption or product yield reported in the statistics. For the plants not included in the preceding data sources, we calculate the emission by using "top-down method" and allocate the emission with proxies, such as GDP and population. (Page 5, Line 6-13; Page 6, Line 14-18; Page 7, Line 25-26)

(13) Page 5, Line 5
Is there no information on control technologies for boilers?
Response: We do have information on control technologies for boilers. This sentence was revised as: "For industrial boilers, we obtained the location, fuel use amount, and control technologies of over 8 thousand industrial boilers…" (Page 5, line 16-19)

(14) Page 5, Line 9
Is the expression "emission factor method" appropriate? The unit-based approach also uses emission factors. I think it is usually called as "top-down method" (but sometimes confused with top-down estimates utilizing observations including satellites).
Response: "emission factor method" was replaced with "top-down method" in this sentence. (Page 6, Line 7)

(15) Page 5, Line 17
How about speciation of PM2.5 and NMVOCs for unit-based emissions?
Response: As is described in the manuscript, the speciation of $PM_{2.5}$ is from Fu et al. (2013). The $PM_{2.5}$ speciation profile of major sectors is shown in Fig. R2 (Fig. S4 in the manuscript). The speciation of NMVOCs is updated by Wu et al. (2017). The speciation profiles used in the unit-based inventory is the same as those used in the proxy-based inventory. The manuscript is revised accordingly. (Page 6, Line 18-20)

[Figure]

Fig. R2 PM$_{2.5}$ speciation profile of major sectors

(16) Page 5, Line 20 – Page 6, Line 18
References for models and modules are required.
Response: References are added to the manuscript. The revised text is shown as follows: In this work, we use CMAQ version 5.0.2 (EPA, 2014) to simulate the concentration of pollutants. (Page 6, Line 23-24) The Carbon Bond 05 (CB05) and AERO6 (Sarwar et al., 2011) are chosen as the gas-phase and aerosol chemical mechanisms, respectively. (Page 7, Line 5-7)
We use the Weather Research and Forecasting (WRF) model version 3.7.1 (Skamarock et al., 2008) to simulate the meteorological fields. The physics options for the WRF simulation are the Kain-Fritsch cumulus scheme (Kain, 2004), the Morrison double-moment scheme for cloud microphysics (Morrison et al., 2005), the Pleim-Xiu land surface model (Xiu and Pleim, 2001), Pleim-Xiu surface layer scheme (Pleim, 2006), ACM2 (Pleim) boundary layer parameterization (Pleim, 2007), and Rapid Radiative Transfer Model for GCMs radiation scheme (Mlawer et al., 1997). (Page 7, Line 9-14)

(17) Page 6, Line 10
What are "other" configurations? Please show explicitly.
Response: "Other" configurations means the initial and boundary conditions. The meteorological initial and boundary conditions are generated from the Final Operational Global Analysis data (ds083.2) of the National Center for Environmental Prediction (NCEP) at a 1.0º × 1.0º and 6-h resolutions. Default profile data is used for chemical initial and boundary conditions. It is revised accordingly in the manuscript. (Page 7, Line 14-17)

(18) Page 6, Lines 21-23
Is CO not included in this study? Why?
Response: The ambient CO pollution is not a serious issue in China currently. According to China National Environmental Monitoring Centre (data source: http://106.37.208.233:20035/), the daily CO concentration in the BTH region is less than 1.5 mg/m$^3$, which is much lower than the national ambient air quality standard (4 mg/m$^3$). In addition, the influence of CO emission on the formation of PM$_{2.5}$ and O$_3$ is quite small. For these two reasons, we did not include CO emission in this study.
In the model simulations described in this paper, we used CO emissions developed by Janssens-Maenhout et al. (2015).

(19) Page 7 Lines 1-22
Area names are mentioned in these paragraphs. However, horizontal distributions firstly appear later in Fig. 3. Its description should appear before descriptions of areas.
Response: The sequence of these sentences has been adjusted. (Page 8, Line 12-13)

(20) Page 7, Line 6

It is impossible to see many industrial boilers in Fig. 2.

Response: The link was wrong and we revised it to "Fig. 3". (Page 8, Line 12-13)

(21) Page 8, Line 9

I think that NMB and NME are not appropriate metrics in terms of this study. The target of this study is accurate horizontal distributions. However, overestimation in one areas and underestimation in other areas could be cancelled out in these metrics. It is necessary to appropriate metrics which can properly shows improvements realized in this study.

Response: Thank you for this valuable comment.

While the overestimation and underestimation in different areas could be cancelled out in normalized mean bias (NMB), they cannot be cancelled out in the normalized mean error (NME), which characterizes the absolute difference between observation and simulation. Similarly, mean fractional error (MFE) is also an index that will not cancel out the overestimation and underestimation. The NME and MFE for $SO_2$, $NO_2$, $PM_{2.5}$, and $O_3$ are mostly lower with the unit-based inventory than with the proxy-based inventory, which means that the spatial distributions of these pollutants are better captured using the unit-based inventory. (Page 12, Line 5-8)

A major difference between the proxy-based and unit-based inventories is that the traditional proxy-based inventory allocates more emission to the urban area, whereas the unit-based inventory allocates more emission to suburban area where more factories are located. To quantify the impact of changed emission distribution between urban and suburban areas, we introduced the metric of "concentration gradient", which is defined as the ratios of urban concentrations to suburban concentrations. The concentration gradients simulated with the unit-based inventory agree much better with observations than those simulated with the proxy-based inventory, implying that the unit-based emission inventory better reproduces the distributions of pollutant emissions between the urban and suburban areas. (Page 12, Line 5-22)

In addition, most of the observational sites (70 out of 80) are located in urban area. (Page 9, Line 19-20) Therefore, the calculated NMB is dominated by the behavior of the urban sites, and is not likely to be significantly cancelled out by the limited suburban sites.

Finally, we have shown the model performance for major air pollutants at each individual site in Beijing in the revised Supplementary Information (Table S6-S9). For the urban sites, the concentrations of $PM_{2.5}$, $SO_2$ and $NO_2$ are much lower with the unit-based inventory than with the proxy-based inventory. For the suburban sites, however, the concentrations are either slightly higher or slightly lower with the unit-based inventory than with the proxy-based inventory. The situation for ozone is quite the opposite. The ozone concentration at urban sites is higher with the unit-based inventory than with the proxy-based inventory. In suburban sites, it is lower with the unit-based inventory than with the proxy-based inventory. In addition, for the simulations with the unit-based inventory, the NME and MFE of individual sites are usually lower than those with the proxy-based inventory while the correlation efficient is usually higher, which means that the error is generally smaller and the trend is more

similar to the observation when the unit-based inventory is used.

(22) Page 8, Lines 15-17

What is a possible reason for the poor model performance on SO2?

Response: The overestimation of $SO_2$ concentrations may be due to the lack of several $SO_2$ reaction mechanisms in CMAQ, such as heterogeneous reactions of $SO_2$ on the surface of dust particles (Fu et al., 2016), the oxidation of $SO_2$ by NOx in aerosol liquid water (Cheng et al., 2016;Wang et al., 2016a), the effects of $SO_2$ and $NH_3$ on secondary organic aerosol formation (Chu et al., 2016), etc.

The biased spatial distribution of $SO_2$ emissions from residential combustion may also contribute to the overestimation. A large fraction of residential combustion takes place in the rural areas. In this work, however, the emission of residential combustion is allocated by GDP and population, which leads to an overestimation of $SO_2$ emission in urban area and hence an overestimation of $SO_2$ concentration. (Page 10, Line 4-11)

(23) Page 9, Lines 19-20

I cannot find any descriptions on plume rise before here. How to gather stack information? How to calculate plume rise? These descriptions are required in the method section.

Response: The stack information required for plume rise calculation includes stack height, flue gas temperature, chimney diameter and flue gas velocity. For power plants, we get the stack height from Compilation of power industry statistics (China Electricity Council, 2015). For the stack height of cement factories, we refer to the emission standard of air pollutants for cement industry (Ministry of Environmental Protection of China, 2013). For the stack height of glass, brick, lime and ceramics industries, we refer to emission standard of air pollutants for industrial kiln and furnace (Ministry of Environmental Protection of China, 1997). For the stack height of non-ferrous metal smelter, coking, refinery and chemical industries, as well as the flue gas temperature, chimney diameter and flue gas velocity for all industrial sectors, we refer to the national information platform of pollutant discharge permit (http://114.251.10.126/permitExt/outside/default.jsp), where we can find very detailed information of the plants with the pollutant discharge permit. For the sources without the pollutant discharge permit, we use the parameters of the plant with a similar production output or coal consumption. (Page 5, Line 23 to Page 6, Line 5) The data source of stack information is shown in Table R4 (Table S5 in the manuscript).

Table R4 Data source of stack information

| Sector | Stack height | Flue gas temperature, Chimney diameter, Flue gas velocity |
|---|---|---|
| Power plant | Compilation of power industry statistics | National information platform of pollutant discharge permit |
| Cement plant | Emission standard of air pollutants for cement industry | |

| Glass, brick, lime and ceramics industries | Emission standard of air pollutants for industrial kiln and furnace |
| --- | --- |
| Non-ferrous metal smelter, coking, refinery and chemical industries | National information platform of pollutant discharge permit |

Plume rise is calculated with a built-in algorithm of CMAQ based on the Briggs's scheme (Briggs, 1982). In this algorithm, plume rise is estimated by simulating the buoyancy effect and momentum rise, using hourly and gridded meteorological data. Then, the plume is distributed into the vertical layers that the plume intersects based on the pressure in each layer. (Page 5, Line 20-22; Page 8, Line 3-5)

(24) Page 10, Line 1
Details of "concentration gradient" are necessary. How to select urban and suburban locations? Are monthly mean concentrations used?
Response: For Beijing, the suburban areas refer to the districts that are far from the center of the city (the red star in Fig. 2). From Fig. 2 we can see that there are 8 sites located in the urban districts in Beijing. In the north, there are four sites far away from the city center and close to the city border. We treat the four sites in the north as suburban sites and the others as urban sites. For Tianjin, as shown in Fig. 3, there are two city centers. Ten sites are located in urban area and 5 sites are located in suburban area. In the calculation of the concentration gradient, monthly mean concentrations are used (Page 12, Line 10). These figures are added to the Supplementary Information. (Fig. S4-S5)

[Figure]

Fig. R3 The observational sites in Beijing

[Figure]

Fig. R4 The observational sites in Tianjin

(25) Page 10, Lines 24-27

I think it is not enough to explain changes of NO3- only by NOx sensitivities. I do not think they are main reasons. SO42- concentrations in the unit-based approach are much lower than the proxy-based approach whereas NH4+ is almost constant as shown in Fig.

7. In this case, more HNO3 is converted to NO3- with excess NH4+ whereas these processes depend on abundance of HNO3 or NH3.

Response: Thank you for the valuable idea. We have added this reason to explain the changes of $NO_3^-$ as follows:

As for nitrate, concentration of nitrate in the simulation with unit-based inventory is much higher than that with proxy-based inventory in winter while the differences between the results with two inventories vary with location in summer. Sulfate concentrations in the unit-based approach are much lower than the proxy-based approach. In this case, more abundant $NH_3$ is available to react with $HNO_3$, leading to enhanced formation of $NO_3^-$. (Page 13, Line 11-14)

[revised manuscript text omitted]

---

## Author Comment (AC2) · 28 Jan 2019

Reviewer 2:

This is a timely paper that describes the development of a unit-based industrial emission inventory in northern China, which still suffers severe air pollution even though the government has put tremendous amount of effort in emission controls. A detailed, united-based emission inventory will be of great value when air quality models are used in developing/assessing emission control strategies. The paper is generally well-written. I would recommend the paper be published in ACP after addressing my comments below.

Response: We appreciate the reviewer's valuable comments which help us improve the quality of the manuscript. We have carefully revised the manuscript according to the reviewers' comments. Point-to-point responses are given below. The original comments are in black, while our responses are in blue.

(1) The paper lacks details on how vertical distribution of point source emissions are treated in the simulation. In the results section, it is mentioned that plume rise contributes to the difference between the CMAQ results. However, no details were provided on how the parameters needed for plume rise calculations are obtained. In my understanding, such data are not universally available (even in the US) so presumably the same situation is applicable in China. What is the criteria for selecting point sources for plume rise calculation and how missing information is estimated. I also believe that the authors should perform off-line emission vertical distribution calculations and compare with the empirical vertical distribution used for the proxy-based emission inventory. For many of people without access to the detailed unit-based emission inventory, it will be useful to see this information so that vertical distribution in the traditional inventories can also be improved.

Response:

In the simulation, the vertical distribution of point source emissions is calculated by employing a built-in plume-rise calculation algorithm of CMAQ based on the Briggs's scheme (Briggs, 1982). In this algorithm, plume rise is estimated by simulating the buoyancy effect and momentum rise, using hourly and gridded meteorological data. Then, the plume is distributed into the vertical layers that the plume intersects based on the pressure in each layer. (Page 5, Line 20-22; Page 8, Line 2-4)

The stack information required for plume rise calculation includes stack height, flue gas temperature, chimney diameter and flue gas velocity. For power plants, we get the stack height from Compilation of power industry statistics (China Electricity Council, 2015). For the stack height of cement factories, we refer to the emission standard of air pollutants for cement industry (Ministry of Environmental Protection of China, 2013). For the stack height of glass, brick, lime and ceramics industries, we refer to emission standard of air pollutants for industrial kiln and furnace (Ministry of Environmental Protection of China, 1997). For the stack height of non-ferrous metal smelter, coking, refinery and chemical industries, as well as the flue gas temperature, chimney diameter and flue gas velocity for all industrial sectors, we refer to the national information platform of pollutant discharge permit (http://114.251.10.126/permitExt/outside/default.jsp), where we can find very detailed information of the plants with the pollutant discharge permit. For the sources without the pollutant discharge permit, we use the parameters of the plant with a similar production output or coal consumption. (Page 5, Line 23 to Page 6, Line 5) The data source of stack information is shown in Table R1 (Table S5 in the manuscript).

Table R1 Data source of stack information

| Sector | Stack height | Flue gas temperature, Chimney diameter, Flue gas velocity |
|---|---|---|
| Power plant | Compilation of power industry statistics | National information platform of pollutant discharge permit |

| Cement plant | Emission standard of air pollutants for cement industry |
|---|---|
| Glass, brick, lime and ceramics industries | Emission standard of air pollutants for industrial kiln and furnace |
| Non-ferrous metal smelter, coking, refinery and chemical industries | National information platform of pollutant discharge permit |

The vertical distribution of emissions after plume rise for each industrial sector is shown in Table R2 (Table S6 in the manuscript). The empirical vertical distribution used for the proxy-based emission inventory is also provided for reference (Table R3, Table S7 in the manuscript). In general, compared with the proxy-based inventory, more emissions are distributed in higher vertical levels in the unit-based inventory with plume rise considered.

Table R2 Vertical distribution of emissions for each industrial sector in the unit-based inventory with plume rise considered

| Layer | Sigma value | Level height (m) | Power plants | | Iron plants | | Cement plants | | Industrial boilers | | Industrial process | |
|---|---|---|---|---|---|---|---|---|---|---|---|---|
| | | | Jan | Jul | Jan | Jul | Jan | Jul | Jan | Jul | Jan | Jul |
| 1 | 0.995 | 35 | 0% | 0% | 0% | 0% | 3% | 3% | 3% | 4% | 3% | 6% |
| 2 | 0.99 | 85 | 0% | 0% | 0% | 0% | 9% | 11% | 21% | 21% | 28% | 31% |
| 3 | 0.98 | 140 | 0% | 0% | 0% | 6% | 26% | 32% | 32% | 41% | 45% | 36% |
| 4 | 0.96 | 210 | 6% | 9% | 60% | 85% | 49% | 49% | 39% | 31% | 20% | 24% |
| 5 | 0.94 | 310 | 15% | 17% | 38% | 9% | 13% | 5% | 5% | 2% | 3% | 3% |
| 6 | 0.91 | 440 | 47% | 45% | 2% | 0% | 1% | 0% | 0% | 0% | 0% | 0% |
| 7 | 0.86 | 610 | 31% | 29% | 0% | 0% | 0% | 0% | 0% | 0% | 0% | 0% |

Table R3 Vertical distribution of emissions for each industrial sector in the proxy-based inventory

| Layer | Sigma value | Level height (m) | Power plants | Iron plants | Cement plants | Industrial boilers | Industrial process |
|---|---|---|---|---|---|---|---|
| 1 | 0.995 | 35 | 0% | 6% | 6% | 50% | 6% |
| 2 | 0.99 | 85 | 10% | 26% | 26% | 30% | 26% |
| 3 | 0.98 | 140 | 10% | 68% | 68% | 20% | 68% |
| 4 | 0.96 | 210 | 30% | 0% | 0% | 0% | 0% |
| 5 | 0.94 | 310 | 20% | 0% | 0% | 0% | 0% |
| 6 | 0.91 | 440 | 20% | 0% | 0% | 0% | 0% |
| 7 | 0.86 | 610 | 10% | 0% | 0% | 0% | 0% |

To separate the contributions of horizontal and vertical distributions to the differences between the simulations using the proxy-based and unit-based inventories, we have conducted an additional simulation in which the unit-based inventory is used but the emission heights are assumed to be the same as the proxy-based inventory. The amount of emission is the same as the other two scenarios. We call the inventory used in this simulation "hypo unit-based inventory".

Fig. R1 (Fig. 5 in the revised manuscript) shows the distribution of the monthly (January and July) mean concentrations of $SO_2$, $NO_2$, ozone, daily maximum 1-h averaged ozone, daily maximum 8-h averaged

ozone and $PM_{2.5}$ simulated with the proxy-based inventory, and the differences between the proxy-based simulation and the other two simulations (Diff1: hypo unit-based minus proxy-based; Diff2: unit-based minus proxy-based). For $SO_2$, $NO_2$ and $PM_{2.5}$, the concentrations in the urban area are generally higher with the proxy-based inventory than those with the unit-based inventory, especially in winter. In January, large concentration differences between simulations with two inventories are found in urban Tianjin, Tangshan, Baoding and Shijiazhuang, where a large amount of industrial emissions is allocated in the proxy-based inventory due to large population density. The simulation of July follows the same pattern but the concentrations and the difference between the concentrations with two inventories are lower than those of January. In some areas where many factories are located, such as the northern part of Xingtai city, the concentration with unit-based inventory is higher because of a high emission intensity. There are two reasons for the difference between results with proxy-based and unit-based inventories. The first one is the spatial distribution. With detailed information of industrial sectors, more emissions are allocated to certain locations in suburban/rural areas in the unit-based emission inventory. From "Diff1" (hypo unit-based minus proxy-based), we can see that the improved horizontal distribution of the unit-based emission inventory significantly decreases the $PM_{2.5}$, $SO_2$, and $NO_2$ concentrations in most urban centers, and significantly increases the concentrations in a large fraction of suburban and rural areas, especially the areas where large industrial plants are located in. The other reason is vertical distribution. Plume rise is calculated in the simulation with the unit-based inventory, which causes the difference of emissions in vertical layers. The higher the pollutants are emitted, the lower the ground concentration becomes. From the differences between Diff1 and Diff2 we can see that the plume rise leads to lower concentrations over the whole region. The results of the additional simulation have been added to the revised manuscript (Page 11, Line 6 to 26; Page 14, Line 2-4)

[Figure]

Fig. R1 Spatial distribution of the monthly (January and July) mean concentrations of $SO_2$, $NO_2$, ozone, daily maximum 1-h averaged ozone, daily maximum 8-h averaged ozone and $PM_{2.5}$ with the proxy-based inventory, and the differences between the other two simulations and proxy-based inventory (Diff1: hypo unit-based minus proxy-based; Diff2: unit-based minus proxy-based). The units are $\mu g/m^3$ for all panels.

(2) One of the major conclusions from the study is that unit-based emission inventory leads to significant improvement in the model performance. However, the only quantitative assessment is monthly average concentrations of SO2, NO2, O3, PM2.5 using all the stations in the domain. This is not sufficient as information is lost in the averaging process. At minimal, the authors should show performance of these pollutants at each individual sites. Time series should also be shown for sites with significant differences. It will help identify the cause of the differences. For O3, it is necessary to show performance of 1-hr peak ozone and 8-hr daily maximum. Very large error still exists for SO2. More discussion of this over-estimation should be included.

Response:

(1) Following the reviewer's comment, we summarize the model performance for major air pollutants at each individual site in Beijing (12 sites out of a total of 80 sites in the BTH region) in Table R1-R4 (Table S8-S11). The time series of $PM_{2.5}$ concentration at representative urban sites (Wanshouxigong and Dongsi) and suburban sites (Huairou and Shunyi) are shown in Fig. R2-R3 (Fig. S7-S8). For the urban sites, the concentrations of $PM_{2.5}$, $SO_2$ and $NO_2$ are much lower with the unit-based inventory than with the proxy-based inventory. For the suburban sites, however, the concentrations are either slightly higher or slightly lower with the unit-based inventory than with the proxy-based inventory. The situation for ozone is quite the opposite. The ozone concentration at urban sites is higher with the unit-based inventory than with the proxy-based inventory. In suburban sites, it is lower with the unit-based inventory than with the proxy-based inventory. In addition, for the simulations with the unit-based inventory, the normalized mean error (NME) and mean fractional error (MFE) of individual sites are usually lower than those with the proxy-based inventory while the correlation efficient is usually higher, which means that the error is generally smaller and the trend is more similar to the observation when the unit-based inventory is used.

(2) The figures of time series of $PM_{2.5}$ concentration corroborates the preceding conclusion. At urban sites, the concentration with the unit-based inventory is substantially lower than that with the proxy-based inventory throughout the simulation periods. For suburban sites, the concentration is slightly lower with the unit-based inventory than that with the proxy-based inventory in most of the simulation period.

(3) To further quantify the impact of changed emission distribution between urban and suburban areas, we introduced the metric of "concentration gradient", which is defined as the ratios of urban concentrations to suburban concentrations. As shown in Fig. 6 in the manuscript, the concentration gradients simulated with the unit-based inventory agree much better with observations than those simulated with the proxy-based inventory, implying that the unit-based emission inventory better reproduces the distributions of pollutant emissions between the urban and suburban areas.

The preceding tables, figures, and descriptions have been added to the revised manuscript. (Page 12, Line 5-22)

(4) For ozone, the performance statistics for 1-hr peak ozone and 8-hr daily maximum concentration have been calculated, which is shown in Table 2 in the manuscript.

(5) The overestimation of $SO_2$ concentrations may be due to the lack of several $SO_2$ reaction mechanisms in CMAQ, such as heterogeneous reactions of $SO_2$ on the surface of dust particles (Fu et al., 2016), the oxidation of $SO_2$ by NOx in aerosol liquid water (Cheng et al., 2016;Wang et al., 2016), the effects of $SO_2$ and $NH_3$ on secondary organic aerosol formation (Chu et al., 2016), etc. The biased

spatial distribution of $SO_2$ emissions from residential combustion may also contribute to the overestimation. A large fraction of residential combustion takes place in the rural areas. In this work, however, the emission of residential combustion is allocated by GDP and population, which leads to an overestimation of $SO_2$ emission in urban area and hence an overestimation of $SO_2$ concentration. (Page 10, Line 4-11)

Table R4 The statistics for model performance of PM$_{2.5}$ with proxy-based and unit-based inventories

| Months | Sites | Concentration (μg/m$^3$) | | | NMB | | NME | | MFB | | MFE | | R | |
|---|---|---|---|---|---|---|---|---|---|---|---|---|---|---|
| | | proxy-based | unit-based | OBS | proxy-based | unit-based | proxy-based | unit-based | proxy-based | unit-based | proxy-based | unit-based | proxy-based | unit-based |
| Jan | Wanshouxigong | 111.2 | 88.0 | 108.4 | 3% | -19% | 54% | 49% | 20% | -3% | 62% | 60% | 0.58 | 0.60 |
| | Dingling Tomb | 36.8 | 36.9 | 69.3 | -47% | -47% | 54% | 53% | -59% | -55% | 72% | 68% | 0.67 | 0.68 |
| | Dongsi | 112.2 | 91.6 | 104.1 | 8% | -12% | 58% | 51% | 24% | 6% | 64% | 61% | 0.55 | 0.56 |
| | Heaven Temple | 110.7 | 90.3 | 97.6 | 13% | -8% | 58% | 51% | 31% | 10% | 64% | 61% | 0.59 | 0.60 |
| | Nongzhanguan | 92.6 | 77.7 | 101.9 | -9% | -24% | 52% | 50% | 4% | -10% | 59% | 58% | 0.57 | 0.59 |
| | Guanyuan | 110.5 | 86.4 | 100.6 | 10% | -14% | 60% | 51% | 23% | 1% | 65% | 61% | 0.55 | 0.56 |
| | Haidian | 86.8 | 70.0 | 109.3 | -21% | -36% | 55% | 53% | -17% | -35% | 66% | 67% | 0.53 | 0.54 |
| | Shunyi | 89.4 | 83.3 | 92.3 | -3% | -10% | 56% | 54% | 8% | 3% | 62% | 61% | 0.55 | 0.55 |
| | Huairou | 49.8 | 48.5 | 86.9 | -43% | -44% | 57% | 55% | -71% | -69% | 86% | 82% | 0.61 | 0.62 |
| | Changping | 74.7 | 70.3 | 85.6 | -13% | -18% | 54% | 51% | -4% | -8% | 58% | 56% | 0.57 | 0.58 |
| | Olympic center | 93.6 | 82.5 | 94.8 | -1% | -13% | 56% | 50% | 9% | 1% | 62% | 59% | 0.57 | 0.59 |
| | Gucheng | 77.0 | 63.2 | 102.0 | -25% | -38% | 50% | 51% | -19% | -37% | 60% | 64% | 0.59 | 0.60 |
| Jul | Wanshouxigong | 55.7 | 50.0 | 96.4 | -42% | -48% | 55% | 58% | -51% | -61% | 69% | 75% | 0.53 | 0.52 |
| | Dingling Tomb | 24.6 | 26.1 | 83.7 | -71% | -69% | 74% | 72% | -106% | -102% | 112% | 109% | 0.55 | 0.57 |
| | Dongsi | 57.3 | 52.2 | 110.0 | -48% | -53% | 57% | 59% | -54% | -61% | 72% | 75% | 0.56 | 0.55 |
| | Heaven Temple | 58.0 | 52.5 | 103.3 | -44% | -49% | 56% | 58% | -52% | -61% | 70% | 74% | 0.51 | 0.50 |
| | Nongzhanguan | 54.4 | 50.2 | 91.7 | -41% | -45% | 54% | 55% | -54% | -59% | 70% | 72% | 0.50 | 0.50 |
| | Guanyuan | 54.8 | 49.8 | 99.6 | -45% | -50% | 55% | 57% | -59% | -67% | 73% | 78% | 0.56 | 0.56 |
| | Haidian | 42.8 | 39.9 | 99.8 | -57% | -60% | 61% | 63% | -88% | -91% | 93% | 95% | 0.60 | 0.60 |
| | Shunyi | 60.2 | 55.8 | 101.7 | -41% | -45% | 54% | 55% | -41% | -47% | 67% | 70% | 0.58 | 0.57 |
| | Huairou | 44.8 | 45.0 | 101.2 | -56% | -56% | 60% | 59% | -89% | -78% | 96% | 85% | 0.68 | 0.68 |
| | Changping | 37.2 | 39.0 | 91.9 | -60% | -58% | 65% | 63% | -77% | -74% | 86% | 84% | 0.65 | 0.67 |
| | Olympic center | 50.5 | 47.1 | 104.8 | -52% | -55% | 57% | 59% | -73% | -77% | 81% | 83% | 0.60 | 0.59 |
| | Gucheng | 38.2 | 36.9 | 97.2 | -61% | -62% | 63% | 64% | -96% | -98% | 99% | 100% | 0.64 | 0.63 |

Table R5 The statistics for model performance of $NO_2$ with proxy-based and unit-based inventories

| Months | Sites | Concentration (µg/m³) | | | NMB | | NME | | MFB | | MFE | | R | |
|---|---|---|---|---|---|---|---|---|---|---|---|---|---|---|
| | | proxy-based | unit-based | OBS | proxy-based | unit-based | proxy-based | unit-based | proxy-based | unit-based | proxy-based | unit-based | proxy-based | unit-based |
| Jan | Wanshouxigong | 135.3 | 96.0 | 81.1 | 67% | 18% | 81% | 47% | 43% | 11% | 59% | 46% | 0.60 | 0.64 |
| | Dingling Tomb | 39.9 | 41.4 | 37.2 | 7% | 11% | 60% | 58% | 4% | 11% | 60% | 60% | 0.64 | 0.66 |
| | Dongsi | 135.1 | 101.4 | 66.3 | 104% | 53% | 116% | 75% | 59% | 37% | 72% | 60% | 0.43 | 0.42 |
| | Heaven Temple | 134.9 | 101.8 | 73.4 | 84% | 39% | 94% | 61% | 48% | 21% | 60% | 50% | 0.57 | 0.58 |
| | Nongzhanguan | 108.0 | 93.9 | 68.0 | 59% | 38% | 79% | 59% | 33% | 26% | 58% | 51% | 0.60 | 0.63 |
| | Guanyuan | 141.7 | 99.9 | 76.1 | 86% | 31% | 103% | 61% | 42% | 12% | 63% | 52% | 0.59 | 0.59 |
| | Haidian | 106.1 | 80.0 | 93.5 | 13% | -14% | 66% | 50% | -12% | -32% | 62% | 60% | 0.47 | 0.48 |
| | Shunyi | 100.7 | 92.7 | 56.8 | 77% | 63% | 93% | 80% | 42% | 37% | 62% | 58% | 0.61 | 0.61 |
| | Huairou | 54.8 | 54.0 | 56.7 | -3% | -5% | 72% | 68% | -45% | -41% | 85% | 81% | 0.55 | 0.57 |
| | Changping | 102.3 | 95.3 | 57.0 | 80% | 67% | 98% | 87% | 39% | 34% | 60% | 57% | 0.54 | 0.56 |
| | Olympic center | 113.9 | 100.0 | 67.7 | 68% | 48% | 86% | 66% | 43% | 37% | 63% | 58% | 0.60 | 0.62 |
| | Gucheng | 95.0 | 73.6 | 75.9 | 25% | -3% | 61% | 46% | 12% | -9% | 55% | 52% | 0.61 | 0.64 |
| Jul | Wanshouxigong | 22.1 | 18.4 | 41.3 | -46% | -56% | 59% | 62% | -72% | -86% | 83% | 92% | 0.17 | 0.15 |
| | Dingling Tomb | 5.7 | 7.2 | 17.1 | -67% | -58% | 70% | 63% | -116% | -102% | 118% | 106% | 0.34 | 0.37 |
| | Dongsi | 25.6 | 22.6 | 43.5 | -41% | -48% | 57% | 58% | -65% | -72% | 78% | 81% | 0.22 | 0.19 |
| | Heaven Temple | 24.5 | 21.0 | 36.4 | -33% | -42% | 56% | 56% | -53% | -63% | 73% | 77% | 0.24 | 0.21 |
| | Nongzhanguan | 22.8 | 21.7 | 44.9 | -49% | -52% | 60% | 60% | -78% | -77% | 87% | 84% | 0.27 | 0.27 |
| | Guanyuan | 21.4 | 18.1 | 42.2 | -49% | -57% | 61% | 62% | -75% | -87% | 86% | 93% | 0.11 | 0.12 |
| | Haidian | 14.9 | 13.7 | 54.3 | -73% | -75% | 74% | 76% | -119% | -123% | 121% | 124% | 0.07 | 0.09 |
| | Shunyi | 21.6 | 19.7 | 28.1 | -23% | -30% | 39% | 41% | -36% | -43% | 54% | 57% | 0.66 | 0.64 |
| | Huairou | 11.9 | 12.8 | 25.0 | -52% | -49% | 62% | 60% | -96% | -86% | 105% | 95% | 0.39 | 0.40 |
| | Changping | 15.1 | 17.1 | 32.5 | -53% | -47% | 58% | 54% | -85% | -75% | 90% | 81% | 0.27 | 0.28 |
| | Olympic center | 20.1 | 18.5 | 48.6 | -59% | -62% | 64% | 64% | -93% | -96% | 97% | 99% | 0.23 | 0.25 |
| | Gucheng | 12.4 | 12.2 | 45.6 | -73% | -73% | 74% | 74% | -118% | -118% | 119% | 118% | 0.23 | 0.24 |

Table R6 The statistics for model performance of $SO_2$ with proxy-based and unit-based inventories

| Months | Sites | Concentration (µg/m³) | | | NMB | | NME | | MFB | | MFE | | R | |
|---|---|---|---|---|---|---|---|---|---|---|---|---|---|---|
| | | proxy-based | unit-based | OBS | proxy-based | unit-based | proxy-based | unit-based | proxy-based | unit-based | proxy-based | unit-based | proxy-based | unit-based |
| Jan | Wanshouxigong | 102.2 | 93.4 | 62.0 | 65% | 51% | 77% | 67% | 60% | 51% | 69% | 65% | 0.51 | 0.51 |
| | Dingling Tomb | 36.0 | 36.7 | 35.4 | 2% | 3% | 73% | 71% | -9% | -4% | 70% | 69% | 0.43 | 0.44 |
| | Dongsi | 99.8 | 91.8 | 56.7 | 76% | 62% | 86% | 75% | 64% | 57% | 72% | 67% | 0.52 | 0.53 |
| | Heaven Temple | 99.7 | 92.0 | 47.1 | 112% | 95% | 124% | 112% | 81% | 74% | 88% | 84% | 0.30 | 0.30 |
| | Nongzhanguan | 88.0 | 82.7 | 58.9 | 50% | 40% | 69% | 62% | 49% | 45% | 64% | 61% | 0.50 | 0.52 |
| | Guanyuan | 101.2 | 91.6 | 54.8 | 85% | 67% | 97% | 84% | 67% | 58% | 77% | 72% | 0.51 | 0.51 |
| | Haidian | 89.7 | 81.7 | 58.2 | 54% | 40% | 81% | 73% | 48% | 40% | 72% | 70% | 0.47 | 0.46 |
| | Shunyi | 68.9 | 66.1 | 44.0 | 57% | 50% | 79% | 75% | 56% | 53% | 73% | 71% | 0.48 | 0.47 |
| | Huairou | 46.3 | 46.5 | 45.6 | 2% | 2% | 66% | 63% | -8% | -4% | 74% | 71% | 0.40 | 0.40 |
| | Changping | 66.3 | 64.0 | 57.6 | 15% | 11% | 58% | 56% | 12% | 8% | 56% | 56% | 0.46 | 0.46 |
| | Olympic center | 87.2 | 82.6 | 58.3 | 50% | 42% | 72% | 66% | 45% | 42% | 65% | 62% | 0.50 | 0.50 |
| | Gucheng | 80.6 | 72.4 | 52.9 | 52% | 37% | 79% | 69% | 47% | 38% | 71% | 66% | 0.52 | 0.52 |
| Jul | Wanshouxigong | 69.9 | 66.5 | 5.6 | 1144% | 1083% | 1168% | 1110% | 149% | 145% | 156% | 153% | -0.31 | -0.31 |
| | Dingling Tomb | 9.7 | 10.5 | 4.6 | 112% | 128% | 168% | 178% | 52% | 58% | 97% | 98% | 0.12 | 0.11 |
| | Dongsi | 74.6 | 71.8 | 9.6 | 680% | 650% | 696% | 669% | 135% | 133% | 141% | 141% | -0.05 | -0.06 |
| | Heaven Temple | 67.3 | 64.0 | 7.4 | 805% | 762% | 831% | 790% | 136% | 133% | 146% | 144% | -0.27 | -0.28 |
| | Nongzhanguan | 62.9 | 61.0 | 8.7 | 622% | 600% | 649% | 627% | 127% | 128% | 137% | 138% | -0.07 | -0.08 |
| | Guanyuan | 77.0 | 73.5 | 8.5 | 802% | 761% | 842% | 802% | 149% | 147% | 155% | 153% | 0.04 | 0.04 |
| | Haidian | 62.5 | 60.2 | 12.2 | 413% | 394% | 444% | 424% | 96% | 95% | 122% | 120% | -0.26 | -0.26 |
| | Shunyi | 36.8 | 33.5 | 6.5 | 463% | 412% | 498% | 454% | 112% | 106% | 130% | 128% | -0.13 | -0.15 |
| | Huairou | 22.9 | 23.7 | 4.5 | 405% | 422% | 435% | 451% | 112% | 119% | 126% | 129% | -0.01 | -0.02 |
| | Changping | 28.2 | 30.6 | 5.4 | 421% | 466% | 457% | 493% | 114% | 122% | 127% | 133% | -0.06 | -0.02 |
| | Olympic center | 64.2 | 61.8 | 5.0 | 1174% | 1127% | 1193% | 1147% | 141% | 140% | 149% | 149% | -0.09 | -0.08 |
| | Gucheng | 44.1 | 43.0 | 5.2 | 741% | 720% | 767% | 744% | 128% | 128% | 140% | 139% | -0.19 | -0.17 |

Table R7 The statistics for model performance of ozone with proxy-based and unit-based inventories

| Months | Sites | Concentration (µg/m³) | | | NMB | | NME | | MFB | | MFE | | R | |
|--------|-------|-------------|-----------|-----|-------------|------------|-------------|------------|-------------|------------|-------------|------------|-------------|------------|
| | | proxy-based | unit-based | OBS | proxy-based | unit-based | proxy-based | unit-based | proxy-based | unit-based | proxy-based | unit-based | proxy-based | unit-based |
| Jan | Wanshouxigong | 11.0 | 14.0 | 12.0 | -8% | 17% | 100% | 115% | -86% | -71% | 153% | 149% | 0.43 | 0.40 |
| | Dingling Tomb | 47.2 | 46.4 | 35.4 | 33% | 31% | 56% | 54% | 20% | 21% | 80% | 78% | 0.60 | 0.61 |
| | Dongsi | 12.0 | 14.7 | 23.3 | -49% | -37% | 71% | 75% | -116% | -106% | 139% | 137% | 0.50 | 0.44 |
| | Heaven Temple | 11.5 | 14.2 | 20.8 | -45% | -32% | 76% | 82% | -108% | -97% | 144% | 143% | 0.46 | 0.41 |
| | Nongzhanguan | 15.9 | 17.5 | 20.9 | -24% | -16% | 67% | 69% | -87% | -80% | 128% | 127% | 0.61 | 0.59 |
| | Guanyuan | 12.6 | 16.1 | 16.0 | -21% | 1% | 82% | 94% | -78% | -62% | 144% | 142% | 0.50 | 0.43 |
| | Haidian | 18.3 | 22.8 | 14.9 | 23% | 54% | 99% | 122% | -48% | -34% | 139% | 138% | 0.53 | 0.42 |
| | Shunyi | 20.4 | 21.3 | 22.9 | -11% | -7% | 57% | 57% | -66% | -59% | 116% | 113% | 0.71 | 0.70 |
| | Huairou | 40.8 | 40.2 | 26.2 | 56% | 53% | 87% | 86% | 13% | 13% | 101% | 100% | 0.53 | 0.52 |
| | Changping | 26.9 | 28.1 | 25.9 | 4% | 9% | 55% | 55% | -34% | -28% | 90% | 89% | 0.65 | 0.65 |
| | Olympic center | 17.2 | 18.5 | 15.6 | 10% | 18% | 84% | 92% | -50% | -47% | 136% | 136% | 0.58 | 0.52 |
| | Gucheng | 20.6 | 25.1 | 31.5 | -35% | -20% | 64% | 65% | -84% | -64% | 112% | 105% | 0.51 | 0.47 |
| Jul | Wanshouxigong | 57.8 | 58.5 | 104.8 | -45% | -44% | 55% | 55% | -89% | -86% | 104% | 102% | 0.63 | 0.62 |
| | Dingling Tomb | 106.6 | 107.4 | 115.2 | -7% | -7% | 40% | 38% | 1% | 1% | 40% | 39% | 0.49 | 0.56 |
| | Dongsi | 55.1 | 55.5 | 92.8 | -41% | -40% | 57% | 57% | -82% | -80% | 104% | 102% | 0.55 | 0.54 |
| | Heaven Temple | 58.6 | 59.0 | 101.3 | -42% | -42% | 55% | 55% | -82% | -78% | 102% | 99% | 0.63 | 0.61 |
| | Nongzhanguan | 63.4 | 63.1 | 107.0 | -41% | -41% | 57% | 57% | -58% | -57% | 99% | 98% | 0.56 | 0.56 |
| | Guanyuan | 56.5 | 57.3 | 98.7 | -43% | -42% | 61% | 61% | -79% | -77% | 107% | 105% | 0.52 | 0.52 |
| | Haidian | 68.5 | 69.3 | 83.4 | -18% | -17% | 58% | 57% | -28% | -26% | 93% | 92% | 0.54 | 0.55 |
| | Shunyi | 70.2 | 71.2 | 101.9 | -31% | -30% | 44% | 44% | -37% | -33% | 63% | 61% | 0.70 | 0.69 |
| | Huairou | 92.8 | 91.1 | 112.0 | -17% | -19% | 41% | 41% | -12% | -14% | 46% | 46% | 0.54 | 0.54 |
| | Changping | 91.4 | 90.1 | 115.3 | -21% | -22% | 44% | 42% | -17% | -18% | 52% | 51% | 0.51 | 0.55 |
| | Olympic center | 64.3 | 64.7 | 90.3 | -29% | -28% | 59% | 59% | -45% | -43% | 99% | 97% | 0.52 | 0.52 |
| | Gucheng | 80.5 | 80.4 | 100.4 | -20% | -20% | 51% | 50% | -15% | -15% | 73% | 72% | 0.58 | 0.59 |

Fig. R2 The PM$_{2.5}$ concentration in January in Beijing (The black, green and red lines represent observation, results with proxy-based and unit-based inventories)

Fig. R3 The PM$_{2.5}$ concentration in July in Beijing

[Figure]

Fig. R4 Spatial distribution of the monthly (January and July) mean concentrations of SO₂, NO₂, ozone, 1h-peak ozone, MDA8 ozone and PM₂.₅ with the proxy-based inventory, and the differences between the other two simulations and proxy-based inventory (Diff1: hypo unit-based minus proxy-based; Diff2: unit-based minus proxy-based). The units are $\mathbf{\mu g}/\boldsymbol{m^3}$ for all panels.

(3) Table 1 shows "annual average" but only January and July simulations were performed. How did you calculate annual average with only two months of simulation?

Response: We revised "annual average" to "two-month average" in the revised manuscript. (Table 1)

References:

Briggs, G. A.: Plume Rise Predictions, in: Lectures on Air Pollution and Environmental Impact Analyses, edited by: Haugen, D. A., American Meteorological Society, Boston, MA, 59-111, 1982.

Cheng, Y., Zheng, G., Wei, C., Mu, Q., Zheng, B., Wang, Z., Gao, M., Zhang, Q., He, K., Carmichael, G., Poschl, U., and Su, H.: Reactive nitrogen chemistry in aerosol water as a source of sulfate during haze events in China, Science Advances, 2, 10.1126/sciadv.1601530, 2016.

China Electricity Council: Compilation of power industry statistics 2014, China Electricity Council, Beijing, 2015.

Chu, B., Zhang, X., Liu, Y., He, H., Sun, Y., Jiang, J., Li, J., and Hao, J.: Synergetic formation of secondary inorganic and organic aerosol: effect of $SO_2$ and $NH_3$ on particle formation and growth, Atmos Chem Phys, 16, 14219-14230, 10.5194/acp-16-14219-2016, 2016.

Fu, X., Wang, S., Chang, X., Cai, S., Xing, J., and Hao, J.: Modeling analysis of secondary inorganic aerosols over China: pollution characteristics, and meteorological and dust impacts, Sci Rep, 6, 35992, 10.1038/srep35992, 2016.

Ministry of Environmental Protection of China: Emission standard of air pollutants for industrial kiln and furnace, Ministry of Environmental Protection of China (MEP), Beijing, 1997.

Ministry of Environmental Protection of China: Emission standard of air pollutants for cement industry, Ministry of Environmental Protection of China (MEP), Beijing, 2013.

Wang, G., Zhang, R., Gomez, M. E., Yang, L., Levy Zamora, M., Hu, M., Lin, Y., Peng, J., Guo, S., Meng, J., Li, J., Cheng, C., Hu, T., Ren, Y., Wang, Y., Gao, J., Cao, J., An, Z., Zhou, W., Li, G., Wang, J., Tian, P., Marrero-Ortiz, W., Secrest, J., Du, Z., Zheng, J., Shang, D., Zeng, L., Shao, M., Wang, W., Huang, Y., Wang, Y., Zhu, Y., Li, Y., Hu, J., Pan, B., Cai, L., Cheng, Y., Ji, Y., Zhang, F., Rosenfeld, D., Liss, P. S., Duce, R. A., Kolb, C. E., and Molina, M. J.: Persistent sulfate formation from London Fog to Chinese haze, Proc Natl Acad Sci U S A, 10.1073/pnas.1616540113, 2016.

---

## Author Response (AR2)

(1) Page 4, Line 5-6

Please show the table number in SI.

Response: We revised "SI" to "Table S4". (Page 4, Line 5-6)

(2) Page 4, Line 17-

This part is still confusing. Iron and steel production are mentioned as an example in the line 18. However, it seems that the explanation of the symbols in the equation (2) is for cement production. Please reconsider rephrasing.

In addition, these sentences have been added.

"The production processes represented by the first and second terms of equation (2) are frequently performed in different enterprises. For example, for cement production, clinker may be produced in one enterprise and subsequently processed in another enterprise, which is very common."

I agree that. However, the equation (2) is for the enterprise j, isn't it? If processes are divided to multiple enterprises, equation should be also divided.

It is not clear that how to apply information of stacks and locations in the equation (2). Are individual information of stacks and locations applied to each m process in the first term and the second term?

Response:

(1) We rephrased the description of the equation. It is revised as follows:

Some industrial sources involve multiple production process, such as iron and steel production and cement production. Taking cement production for example, emissions are calculated by using the following equation:

$$E_{i,j} = \sum_m \left( AK_{j,m} \times EF_{i,m} \times \left(1 - \eta_{i,j,m}\right)\right) + \left(AC_j \times ef_i \times \left(1 - \eta_{i,j}\right)\right) \quad (2)$$

(Page 4, Line 18-21)

(2) Equation (2) is for enterprise j. For each enterprise, we calculate the emission of each production process. Specifically, the total emission of enterprise j is the sum of the emissions of all production processes in that enterprise. If processes are divided to multiple enterprises, the emission will be considered in the calculation of the emission of each individual enterprise. (Page 5, Line 7-9) Therefore, it is not necessary to divide equation (2).

(3) Individual information of stacks is applied to each m process in the first term and the second term. The locations of different processes in the same enterprise are usually assumed to be the same. (Page 6, Line 9-11)

(3) Page 5, Line 20-

How to interpret emission standards and permits for stack parameters?

Response:

We take the emission standard of air pollutants for cement industry (Ministry of Environmental Protection of China, 2013) as an example. As shown in Table R1 (Translation of part of the emission standard), there are specific requirement for the height of the chimney of different facilities with different production capacities. We assume that the factories are built following the standard. Therefore, we can calculate the minimum stack height with the production capacity of each plant.

Table R1 Emission standard of air pollutants for cement industry. (Translation of part of the standard)

| Facility name | Cement kiln | | | | Drying and grinding mill, coal mill and cooler | | | Other facilities |
|---|---|---|---|---|---|---|---|---|
| Production capacity, t/d | <240 | 240~700 | 700~1200 | >1200 | <500 | 500~1000 | >1000 | 3 m higher than the building |
| Minimum acceptable height | 30 | 45 | 60 | 80 | 20 | 25 | 30 | |

We can also get the detailed stack information from the national information platform of pollutant discharge permit (http://114.251.10.126/permitExt/outside/default.jsp). We choose a random coal-fired power plant (ID of the permit: 9137018135347640XF001P) from the website of the platform as an example, as shown in Table R2 (Translation of part of the permit). We can see that the permit includes most of the stack information of the plant.

Table R2 Basic condition of the chimney of a coal-fired power plant. (Translation of part of the permit)

| No. | Chimney Number | LON | LAT | Chimney height (m) | Chimney diameter (m) | Flue gas temperature (℃) |
|---|---|---|---|---|---|---|
| 1 | DA001 | 117°27' | 36°38' | 100 | 3 | 50 |
| 2 | DA004 | 117°27' | 36°38' | 80 | 3 | 50 |
| 3 | DA005 | 117°27' | 36°38' | 80 | 3.4 | 50 |

(4) Page 8, Line 2-

Was plume-in-grid utilized?

Response: No. The focus of this research is to study the influence of horizontal and vertical distribution of emissions from industrial point sources on simulated air quality. The difference between the hypo unit-based inventory and unit-based inventory represents the influence of vertical distribution. If plume-in-grid was utilized, it would be more difficult to isolate the impact of emission distribution because of the chemical reactions in sub-grid scale.

Nonetheless, we recognize that using plume-in-grid might help to further improve the model performance, which merits further in-depth study. (Page 15, Line 1-2)

(5) Page 10, Line 4-

Do the authors think uncertainties in these factors are dominant? Are there no remaining problems in emission inventory for industries? SO2 is overestimated by 100 ug/m3 in winter. If they are converted to PM through chemical processes listed here, overestimation of PM2.5 would become even much larger.

SO2 emission factors are determined by sulfur contents. They should be relatively reliable. I suppose it may not be so easy to obtain accurate removal efficiencies from each enterprise. Do the authors expect little uncertainties in them?

Response: We agree with the reviewer that the overestimation of $SO_2$ concentration may also be due to uncertainty in emission inventory, especially the uncertainty in the removal efficiencies of $SO_2$ control facilities. We have mentioned this possible reason in the revised manuscript. (Page 10, Line 15-17)

In fact, our $SO_2$ emission estimates are already lower than most previous studies. Fig. R1 (Fig.S3 (a) in the manuscript) compares the $SO_2$ emission of BTH region with those calculated in other studies. The $SO_2$ emission in Beijing and Tianjin in this study is much lower than other studies. As for the $SO_2$ emission in Hebei province, the emission in this study is close to other studies. Further studies are needed to determine the reasons for the discrepancy and improve the simulation results.

[Figure]

Fig. R1 Emissions of SO$_2$ compared with other studies.

(6) Page 12, Line 26

Page 13, Line 4

Table 2 -> Table 3

Response: The manuscript is revised accordingly.

(7) SI Page 4, Figure S2

Abbreviations of sectors are not known.

Response: The full names of the sectors are added to the manuscript. (SI Page 4, Figure S2)

References:

Cai, S., Wang, Y., Zhao, B., Wang, S., Chang, X., and Hao, J.: The impact of the "Air Pollution Prevention and Control Action Plan" on PM2.5 concentrations in Jing-Jin-Ji region during 2012-2020, Sci Total Environ, 580, 197-209, 10.1016/j.scitotenv.2016.11.188, 2017.

Li, M., Zhang, Q., Kurokawa, J., Woo, J. H., He, K. B., Lu, Z. F., Ohara, T., Song, Y., Streets, D. G., Carmichael, G. R., Cheng, Y. F., Hong, C. P., Huo, H., Jiang, X. J., Kang, S. C., Liu, F., Su, H., and Zheng, B.: MIX: a mosaic Asian anthropogenic emission inventory under the international collaboration framework of the MICS-Asia and HTAP, Atmos Chem Phys, 17, 935-963, 10.5194/acp-17-935-2017, 2017.

Ministry of Environmental Protection of China: Emission standard of air pollutants for cement industry, Ministry of Environmental Protection of China (MEP), Beijing, 2013.

Qi, J., Zheng, B., Li, M., Yu, F., Chen, C., Liu, F., Zhou, X., Yuan, J., Zhang, Q., and He, K.: A high-resolution air pollutants emission inventory in 2013 for the Beijing-Tianjin-Hebei region, China, Atmos Environ, 170, 156-168, 10.1016/j.atmosenv.2017.09.039, 2017.

Wang, S. X., Zhao, B., Cai, S. Y., Klimont, Z., Nielsen, C. P., Morikawa, T., Woo, J. H., Kim, Y., Fu, X., Xu, J. Y., Hao, J. M., and He, K. B.: Emission trends and mitigation options for air pollutants in East Asia, Atmos Chem Phys, 14, 6571-6603, 10.5194/acp-14-6571-2014, 2014.